# Interactive and unimodal relationships between plant biomass, abiotic factors, and plant diversity in global grasslands
Marie Spohn [1] ✉, Sumanta Bagchi [2], Jonathan D. Bakker [3], Elizabeth T. Borer [4], Clinton Carbutt [5,6], Jane A. Catford [7,8,9], Christopher R. Dickman [10], Nico Eisenhauer [11,12], Anu Eskelinen [11,13], Nicole Hagenah [14], Yann Hautier [15], Sally E. Koerner [16], Kimberly J. Komatsu [16], Lauri Laanisto [17], Ylva Lekberg [18], Jason P. Martina [19], Holly Martinson [20], Meelis Pärtel [21], Pablo L. Peri [22], Anita C. Risch [23], Nicholas G. Smith [24], Carly Stevens [25], G. F. Ciska Veen [26], Risto Virtanen [13], Laura Yahdjian [27], Alyssa L. Young [16], Hillary S. Young [28] & Eric W. Seabloom [4]

Grasslands cover approximately a third of the Earth's land surface and account for about a third of terrestrial carbon storage. Yet, we lack strong predictive models of grassland plant biomass, the primary source of carbon in grasslands. This lack of predictive ability may arise from the assumption of linear relationships between plant biomass and the environment and an underestimation of interactions of environmental variables. Using data from 116 grasslands on six continents, we show unimodal relationships between plant biomass and ecosystem characteristics, such as mean annual precipitation and soil nitrogen. Further, we found that soil nitrogen and plant diversity interacted in their relationships with plant biomass, such that plant diversity and biomass were positively related at low levels of nitrogen and negatively at elevated levels of nitrogen. Our results show that it is critical to account for the interactive and unimodal relationships between plant biomass and several environmental variables to accurately include plant biomass in global vegetation and carbon models.

Grasslands account for about 34% of the global terrestrial organic carbon stock[1,2] and by far the main source of this carbon is plant biomass. Yet it remains unclear how plant biomass in grasslands is related to the combination of climate[3–5], soil characteristics[6], and plant species diversity[7–9]. One important reason for this might be that many previous studies about plant biomass applied only linear models that do not include interactions among environmental factors and plant diversity[10]. While interactions are widely assumed in ecology, they are rarely tested[11,12], despite the fact that these interactions might be very important for making predictions at the global scale.

Aboveground plant biomass (APB), often called aboveground net primary production in grasslands, is widely assumed to be linearly related to mean annual precipitation (MAP), but evidence for this paradigm is not conclusive[13]. Some studies have found a positive linear relationship between MAP and APB in grasslands[3,14]. Other studies have reported that APB increases linearly with MAP at low levels of MAP and plateaus at high levels of MAP[8,15]. In contrast, a global meta-analysis observed a unimodal, i.e., bell-shaped relationship between MAP and APB in grasslands[5]. It seems likely that very high MAP ( > 1500 mm) can also negatively affect plant biomass since it might cause leaching of nutrients from soil[16], soil acidification[17], and

anoxia[18,19]. Thus, it can be hypothesized that APB in grasslands is highest at intermediate levels of MAP and intermediate levels of the aridity index (AI), which is the ratio of MAP and evapotranspiration.

The relationships of APB and other abiotic ecosystem characteristics might also follow a unimodal function, similar to the relationship between APB and MAP, since it can be expected that APB decreases as conditions diverge from optimum growth conditions. For example, APB in grasslands might be positively related to temperature in the low temperature range, but negatively at very high temperatures since they cause high evapotranspiration. Furthermore, clay is beneficial for APB because it can hold and release nutrients; however, very high clay content promotes anoxia in soils and hampers root growth[20]. Thus, the relationship between APB and soil clay content might also follow a unimodal function. Besides, the availability of some nutrients, such as phosphorus, is highest at neutral pH and reduced at acidic and alkaline pH[21,22]. Thus, the relationship between APB and soil pH might therefore also follow a unimodal function. In addition, low nitrogen inputs to ecosystems are usually beneficial for APB in grasslands because most grasslands are nitrogen limited. However, beyond a certain rate of (atmospheric or fertilizer) nitrogen input, biomass decreases

with further inputs due to nitrogen toxicity[23–25]. Thus, there might be a unimodal (quadratic) relationship between APB and nitrogen.

In addition to unimodal, quadratic models, in which a variable interacts with itself (i.e., is multiplied with itself), different explanatory variables can also interact with each other in their relationship with APB. Specifically, MAP and several environmental factors might interact with each other in their relationships with APB, which could modulate the relationship between MAP and APB. For instance, a high soil clay content might enhance anoxia at sites with high MAP[13,20,26], leading to an interaction of MAP and soil clay content in their relationships with APB. In addition, there might be an interaction of MAP and temperature in their relationships with APB since both MAP and temperature affect the soil water content[27].

Plant diversity can influence plant biomass, yet this relationship is still not well understood in natural grasslands[9,28]. A meta-analysis showed that out of 102 relationships between plant diversity and biomass in grasslands, 39 were positive, nine were negative, and 54 were not statistically significant[28]. One reason why many studies observed no significant relationship between APB and diversity might be that APB not only depends on diversity but also on environmental factors, which might obscure the relationship between APB and plant diversity if not taken into account[7,28–30]. Furthermore, plant diversity and environmental factors might interact in their relationships with APB. For instance, nitrogen can reduce the positive relationship between plant diversity and APB in grasslands[31–33]. The relationship between diversity and APB is also affected by plant productivity (which is often related to nitrogen availability), such that the relationship between APB and diversity in global grasslands is positive at low productivity and negative at high productivity[34]. The complex relationships between plant biomass, plant diversity, and nitrogen may not be captured in linear models that do not include interactions.

The objective of the study is to answer four core questions about APB in grasslands:

1. How is APB in grasslands related to precipitation and other abiotic variables? We hypothesize that a quadratic, unimodal model can better describe the relationship between APB and several environmental factors (MAP, AI, soil clay content, soil pH, and atmospheric nitrogen deposition) than a linear, monotonic model.
2. Does MAP interact with other abiotic variables in its relationship with APB in grasslands? We hypothesize that MAP and soil clay content as well as MAP and temperature interact in their relationships with APB.
3. How does plant species diversity contribute to predicting APB in grasslands? We hypothesize that APB can be predicted from plant diversity if the model includes an interaction with soil nitrogen or atmospheric nitrogen deposition.
4. To what extent do different variables (that are related with APB through complex functions) explain variation in APB when considered together?

To address these research questions and test the hypotheses, we collected standardized APB, plant diversity, and environmental data at 116 natural and semi-natural grasslands on six continents (Fig. 1). MAP at the sites varies between 192 and 1877 mm, while mean annual temperature (MAT) varies between −6.6 and 27.3 °C. The sites are a part of the Nutrient Network Global Research Cooperative (https://nutnet.org). The sites did not receive any fertilizer and were not experimentally manipulated at the time of study. At all sites, APB was measured at peak biomass, i.e., at the specific time of the year when aboveground plant biomass is highest. We used linear, monotonic as well as quadratic functions with one or two predictors (with and without interactions) to address the first three research questions and we applied structural equation modeling to address the fourth question. Structural equation modeling allowed us to explore the contributions of multiple variables (which are related with APB through different functions, and that might covary or interact with each other) to explain variability in APB.

## Results

We found that the relationship between APB and MAP followed a quadratic model across all 116 sites ($P < 0.001$, $R^2 = 0.34$; Table 1, Fig. 2A) and across the subset of 55 sites for which data on soil clay content was available ($P < 0.001$, $R^2 = 0.33$; Table 1). Maximum APB was observed at 1138 mm MAP (Fig. 2A and Supplementary Fig. S1A). We also found a significant linear relationship between APB and MAP across the 116 sites (but not across the 55 sites), yet it had a substantially higher AIC than the quadratic model. If we restricted the analysis to sites with relatively low MAP and MAT (for instance, MAP between 260 and 1200 mm and MAT between 3 and 22 °C), we observed a significant positive linear relationship between APB and MAP ($P < 0.001$, $R^2 = 0.16$, $N = 74$; Supplementary Fig. S2). The relationships between APB and the mean precipitation of the wettest quarter of the year (Pwet) were similar to the relationship between APB and MAP

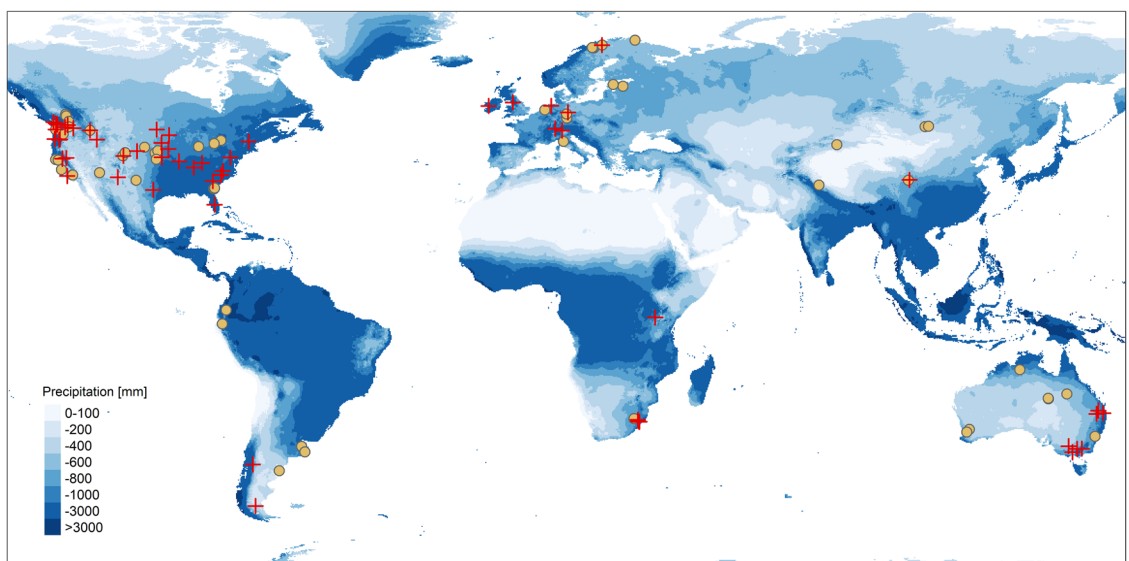

**Fig. 1 | Map depicting mean annual precipitation and the locations of the 116 grassland sites studied here.** Yellow dots indicate the location of the grassland sites for which data on climate, aboveground plant biomass (APB), and plant diversity were collected. Red crosses indicate the 55 sites for which additional data on soil properties, including soil clay content were collected. The map was created in R (version 4.2.2) using the package 'tmap' (version 3.3.3). Mean annual precipitation data were retrieved from Chelsa[65].

**Table 1 | Summary of linear and quadratic models of log-transformed aboveground plant biomass (APB) as a function of one abiotic or biotic predictor (X)**

| Sites | X | | Linear model | | | | Quadratic model | | | |
|---|---|---|---|---|---|---|---|---|---|---|
| | | P value | Multiple $R^2$ | Adjusted $R^2$ | Slope | AIC | P value | Multiple $R^2$ | Adjusted $R^2$ | AIC |
| All sites (N = 116) | MAP | <0.001 | 0.18 | 0.17 | pos. | 263 | <0.001 | 0.34 | 0.33 | 240 |
| | MAT | 0.096 | - | - | - | 283 | 0.241 | - | - | 285 |
| | AI | 0.005 | 0.07 | 0.06 | pos. | 278 | <0.001 | 0.19 | 0.18 | 263 |
| | PWet | <0.001 | 0.09 | 0.08 | pos. | 275 | <0.001 | 0.31 | 0.30 | 245 |
| | TDry | 0.401 | - | - | - | 285 | 0.221 | - | - | 285 |
| | TWarm | 0.531 | - | - | - | 286 | 0.787 | - | - | 288 |
| | TMax | 0.869 | - | - | - | 286 | 0.944 | - | - | 288 |
| | elevation | 0.122 | - | - | - | 284 | 0.017 | 0.07 | 0.05 | 280 |
| | Shannon | 0.250 | - | - | - | 285 | 0.445 | - | - | 287 |
| | Simpson | 0.207 | - | - | - | 284 | 0.440 | - | - | 286 |
| | richness | 0.101 | - | - | - | 283 | 0.261 | - | - | 285 |
| | evenness | 0.656 | - | - | - | 286 | 0.462 | - | - | 286 |
| Subset of sites with soil data (N = 55) | MAP | 0.096 | - | - | - | 124 | <0.001 | 0.33 | 0.30 | 107 |
| | MAT | 0.211 | - | - | - | 125 | 0.320 | - | - | 127 |
| | AI | 0.218 | - | - | - | 125 | <0.001 | 0.24 | 0.21 | 114 |
| | PWet | 0.500 | - | - | - | 126 | <0.001 | 0.26 | 0.20 | 113 |
| | TDry | 0.100 | - | - | - | 124 | 0.122 | - | - | 125 |
| | TWarm | 0.457 | - | - | - | 126 | 0.614 | - | - | 128 |
| | TMax | 0.864 | - | - | - | 127 | 0.977 | - | - | 129 |
| | elevation | 0.203 | - | - | - | 125 | 0.084 | - | - | 124 |
| | NDep | 0.001 | 0.18 | 0.17 | pos. | 116 | 0.002 | 0.21 | 0.18 | 116 |
| | Clay | 0.005 | 0.14 | 0.12 | pos. | 118 | 0.018 | 0.14 | 0.11 | 120 |
| | pH | 0.299 | - | - | - | 126 | 0.544 | - | - | 128 |
| | N | 0.326 | - | - | - | 126 | <0.001 | 0.30 | 0.27 | 110 |
| | C:N | 0.855 | - | - | - | 127 | 0.976 | - | - | 129 |
| | P | 0.105 | - | - | - | 124 | 0.261 | - | - | 126 |
| | K | 0.950 | - | - | - | 127 | 0.967 | - | - | 129 |
| | Ca | 0.041 | 0.08 | 0.06 | pos. | 122 | 0.122 | - | - | 125 |
| | Shannon | 0.697 | - | - | - | 127 | 0.481 | - | - | 127 |
| | Simpson | 0.452 | - | - | - | 126 | 0.638 | - | - | 128 |
| | richness | 0.448 | - | - | - | 126 | 0.626 | - | - | 128 |
| | evenness | 0.966 | - | - | - | 127 | 0.611 | - | - | 128 |

Shown are the P values, the coefficients of determination (multiple and adjusted $R^2$), as well as the Akaike Information Criterion (AIC). Multiple and adjusted $R^2$ are shown for all significant models ($P < 0.05$). For all significant linear models, it is indicated if the regression is positive (pos.) or negative (neg.). All models were calculated based on site-level data. The independent variables (X) were centered, and APB was log-transformed (natural logarithm).

MAP mean annual precipitation, MAT mean annual temperature, AI aridity index, PWet mean precipitation of the wettest quarter of the year, TDry mean temperature of the driest quarter of the year, TWarm mean temperature of warmest quarter of the year, TMax maximum temperature of warmest month, NDep atmospheric nitrogen deposition, Clay soil clay content, pH soil pH, N soil total nitrogen content, C:N soil carbon:nitrogen ratio, P plant available soil phosphorus, K plant available soil potassium, Ca plant available soil calcium.

(Table 1). Furthermore, we found that a quadratic function described the relationship between APB and AI substantially better than a linear function across all 116 sites, as indicated by the considerably lower AIC of the quadratic model (Table 1). Across the subset of 55 sites, we found only a significant quadratic relationship between APB and AI ($P < 0.001, R^2 = 0.24$) but not a significant linear relationship (Table 1). Furthermore, the relationship between APB and elevation of the 116 sites only followed significantly a quadratic, unimodal ($P = 0.017; R^2 = 0.07$) but not a linear, monotonic function (Table 1).

The relationship between APB and soil total nitrogen content also followed a significant quadratic function ($P < 0.001; R^2 = 0.30$; Table 1, Fig. 2B), but not a linear, monotonic one ($P = 0.326$; Table 1). APB had its maximum at a soil nitrogen content of 5.96 g kg$^{-1}$ (Supplementary Fig. 2B and Fig. S1B). Soil nitrogen and MAP were linearly positively correlated (Fig. 3). The relationship of APB and atmospheric nitrogen deposition followed both a significant linear ($P = 0.001, R^2 = 0.18$) and a quadratic model ($P = 0.002, R^2 = 0.21$) with the same AIC (Table 1). Similarly, we found both a significant linear ($P = 0.005, R^2 = 0.14$; Table 1) and a quadratic relationship ($P = 0.018, R^2 = 0.14$; Table 1) for APB as a function of the soil clay content. Plant available soil calcium content was linearly positively related with APB ($P = 0.041, R^2 = 0.08$; Table 1). Other abiotic site characteristics (including soil pH, nutrients, and different measures of temperature, including MAT) as well as plant diversity (Shannon and Simpson indices), plant species richness and evenness were not significantly related with APB, neither through a linear nor a quadratic relationship (Table 1). Of all variables, the Shannon index of plant diversity was only significantly correlated with elevation (Fig. 3). Taken together, we found that quadratic, unimodal functions described the relationship between APB and MAP, APB and AI as well as APB and soil nitrogen better than linear, monotonic functions, as indicated by substantially lower AICs of the quadratic models (Table 1).

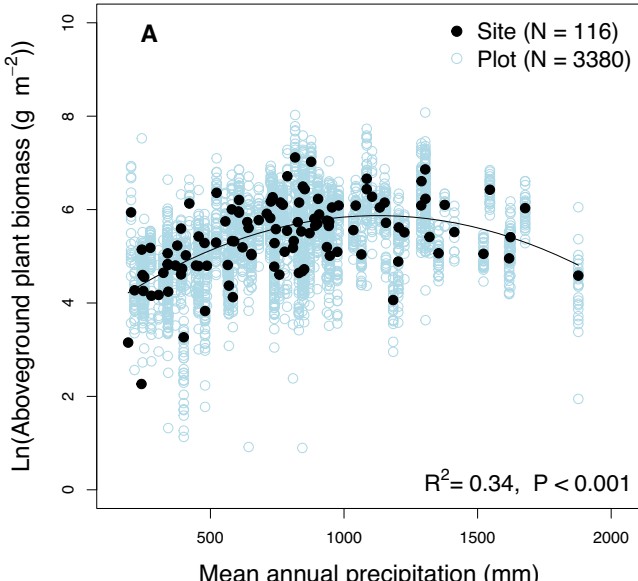

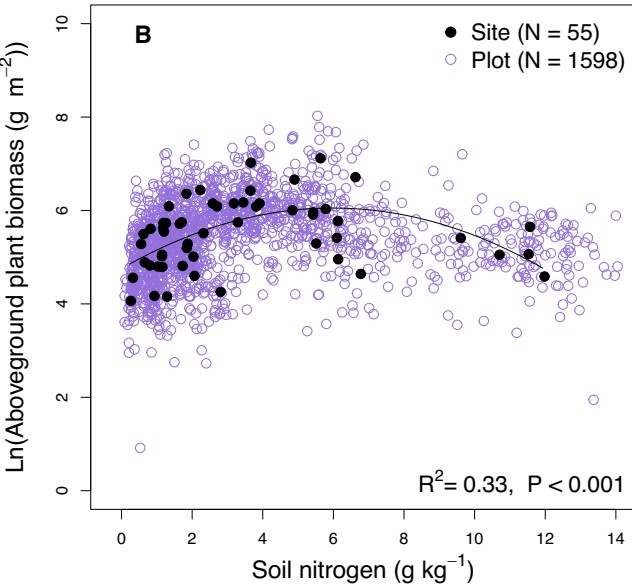

**Fig. 2 | Log-transformed aboveground plant biomass (APB) as a function of mean annual precipitation and soil total nitrogen.** Precipitation ranges from 192 to 1877 mm ($N = 116$; **A**) and the soil total nitrogen content ranges from a site mean of 0.2 to 12.0 g kg$^{-1}$ ($N = 55$; **B**). The quadratic models were calculated based on the site-level data (and not the plot-level data, which are shown to give insight into the variability). APB was log-transformed using the natural logarithm.

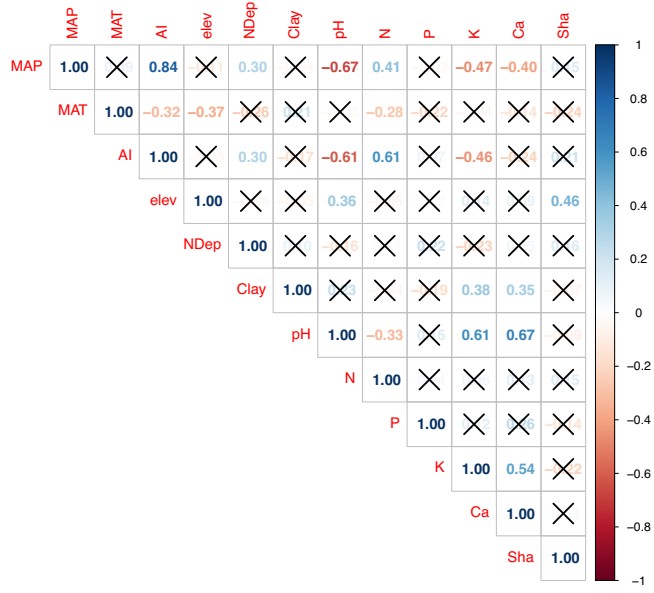

**Fig. 3 | Correlation matrix of log-transformed variables.** The correlation matix shows the Pearson correlation coefficient ($r$) for climate and soil variables as well as the Shannon index of plant diversity (Sha). Correlations with $P > 0.05$ are marked with a cross. Correlations were calculated based on site-level data ($N = 55$). All variables were log-transformed (natural logarithm) before correlations were caclulated. MAP Mean annual precipitation, MAT Mean annual temperature, AI aridity index, elev elevation, NDep atmospheric nitrogen deposition, Clay soil clay content, pH soil pH, N soil total nitrogen content, P plant available soil phosphorus, K plant available soil potassium, Ca plant available soil calcium, Sha Shannon index of plant diversity.

We compared quadratic models of APB for different pairs of variables (Supplementary Table S1), selected according to our second and third hypotheses. The models with the lowest AIC for different pairs of predictors all contained only one quadratic function (Table 2 and S1, Fig. 4), except for the model describing APB as a function of MAP and the mean temperature during the driest quarter of the year (TDry). For three pairs of predictors (MAP and MAT, soil nitrogen and the Shannon index, as well as atmospheric nitrogen deposition and the Shannon index), the two predictors interacted significantly to influence APB (Table 2 and S1, Fig. 4). However, there was no significant interaction of MAP and soil clay content in their relationships with APB (Table 2 and S1, Fig. 4).

Concerning the interaction of soil nitrogen and plant diversity, we found that APB was highest at low plant diversity and medium soil nitrogen

content (Fig. 4C). This relationship was very similar for the Shannon index, the Simpson index, and plant species richness (Supplementary Table S1). APB increased with increasing plant diversity at low levels of soil nitrogen and decreased with increasing plant diversity at medium levels of soil nitrogen (4–7 g nitrogen kg$^{-1}$ soil; Fig. 4C). At low levels of diversity, APB increased strongly with soil nitrogen up to nitrogen contents of about 6 g kg$^{-1}$, while it increased less with nitrogen at high levels of plant diversity (Fig. 4C). Similarly, the best model of APB as a function of the Shannon index and atmospheric nitrogen deposition indicates that APB increased strongly with atmospheric nitrogen deposition at low levels of plant diversity, and less so at very high levels of plant diversity (Table 2 and Fig. 4D). The significant interactions were not caused by the log-transformation of the variable APB, as the same interactions were statistically significant ($P < 0.05$) if the models were calculated with non-transformed data (Supplementary Table S2).

We further tested how different ecosystem characteristics and their interactions act in concert in a piecewise structural equation model (SEM) containing two quadratic functions and two interactions (Fig. 5). The SEM suggests that MAP affected APB directly (through a quadratic function), but mainly indirectly through its significant positive impact on soil nitrogen (see thickness of the arrows depicting the path coefficients in Fig. 5). Soil nitrogen was highly significantly ($P < 0.001$) related with APB through a quadratic function (Fig. 5). Soil nitrogen and the Shannon index significantly interacted to influence APB. Further, atmospheric nitrogen deposition was positively related with APB. MAT and MAP did not significantly interact to influence APB in the SEM (Fig. 5).

## Discussion

Here we show that the relationships of APB with several abiotic ecosystem properties, including MAP, AI, and soil nitrogen, followed unimodal functions, and that plant diversity interacted with soil nitrogen and atmospheric nitrogen deposition in its relationship with APB in global grasslands.

**Table 2 | Summary of the best-fit models of log-transformed aboveground plant biomass (APB) with one or two quadratic terms, without and with multiplicative interactions**

| $X_1$ | $X_2$ | Model structure | P value | P value interaction $X_1$ and $X_2$ | P value interaction $X_1^2$ and $X_2$ | Multiple $R^2$ | Adjusted $R^2$ |
|---|---|---|---|---|---|---|---|
| MAP | MAT | $Y \sim X_1*X_2 + X_1^2*X_2$ | < 0.001 | 0.079 | 0.034 | 0.42 | 0.36 |
| MAP | TDry | $Y \sim X_1 + X_2 + X_1^2 + X_2^2$ | < 0.001 | - | - | 0.41 | 0.36 |
| MAP | Clay | $Y \sim X_1 + X_1^2 + X_2$ | < 0.001 | - | - | 0.38 | 0.34 |
| N | Sha | $Y \sim X_1*X_2 + X_1^2*X_2$ | < 0.001 | 0.027 | 0.025 | 0.38 | 0.32 |
| NDep | Sha | $Y \sim X_1*X_2 + X_1^2*X_2$ | 0.004 | 0.122 | 0.030 | 0.29 | 0.21 |

Shown is the best-fit model for each set of predictors, selected based on the AIC (for model comparisons see Supplementary Table S1). Depicted are the P values of the models and the coefficients of determination (multiple and adjusted $R^2$). For the models with interactions, the P values of the interactions are given. All models were calculated based on site-level data for all sites with data on soil clay content ($N = 55$). The independent variables (X) were centered and APB was log-transformed (natural logarithm).
*MAP* mean annual precipitation, *MAT* Mean annual temperature, *TDry* mean temperature of the driest quarter of the year, *Clay* soil clay content, *N* soil total nitrogen content, *Sha* Shannon index of plant diversity, *NDep* atmospheric nitrogen deposition.

## Biomass as a function of climate

In accordance with our first hypothesis, the relationship between APB and MAP followed a quadratic model. Our results show that APB increased with MAP at low to medium MAP, but decreased with increasing MAP beyond 1138 mm MAP (Fig. 2A). The quadratic, unimodal function describing the relationship between APB and MAP is in accordance with Sun et al. [5] who also found a unimodal relationship between APB and MAP in a global meta-analysis considering grasslands with MAP of up to 2000 mm. Our results show that the relationship between APB and MAP is range-dependent. When we restricted our analysis to sites with relatively low MAP, we found a positive linear relationship. For instance, if we selected sites that match the MAP and MAT levels of sites considered in the seminal paper by Sala et al. [3], i.e., MAP between 260 and 1200 mm and MAT between 3 and 22 °C, we also observed a significant, positive linear relationship between APB and MAP (Supplementary Fig. S2). Thus, our results do not contradict studies reporting a linear, positive relationship between APB and MAP for sites with MAP below 1100 mm [3,14]. Instead, our findings demonstrate that the relationship between APB and MAP is range-dependent, and that APB is only negatively related to MAP beyond 1140 mm (Supplementary Fig. S1A). Some previous studies about the relationship between APB and MAP in grasslands reported that APB increases linearly with MAP at low MAP and plateaued at very high MAP [4,15]. The reason why these studies found no [4] or only a moderate [15] decrease in APB at high MAP is likely that they included very few observations of sites with MAP beyond 1000 mm [15] or beyond 1500 mm [4] in their analyses. These comparisons show that measurements of plant biomass across wide environmental gradients are required to fully understand the relationships among abiotic and biotic ecosystem properties.

According to Whittaker's classification of biomes, all biomes occur under specific combinations of MAT and MAP [35]. For instance, temperate grasslands occur at a MAT of - 3 to 22 °C and MAP below 800 mm. At the same temperature range, but higher MAP, forests dominate [35]. The comparison to Whittaker's classification emphasizes the fact that some of the grasslands studied here are located in areas that would naturally mostly be covered by forest. Yet, other grasslands studied here that have very high MAP are natural grasslands located at relatively high elevation and some of them regularly experience fire, which maintains them as grasslands and prevents bush encroachment [36]. The relatively low APB at sites at high elevation with high MAP might also explain the unimodal relationship between APB and elevation (Table 1). Our findings show that APB of grasslands at high MAP, i.e., at the transition of grasslands and forests, is lower than at intermediate levels of MAP at which grasslands occur. Thus, our results indicate that APB of grasslands does not increase continuously with increasing MAP at the transition from grassland to forest. Instead, APB in grasslands with very high MAP ( > 1500 mm) is reduced compared to APB at the optimal MAP conditions at which grasslands occur at a global scale. Consequently, our study suggests that models assuming a linear positive relationship between plant biomass and MAP at the global scale will likely overestimate grassland biomass at many sites with high precipitation.

Low APB at low MAP is most likely caused by water limitation [3–5]. In contrast, the reason for the negative relationship between APB and MAP at high levels of MAP is likely that high MAP leads to leaching of nutrients that are very mobile in soil, such as potassium [16,37]. This is supported by the negative correlation of MAP and plant available soil potassium and calcium (Fig. 3). In addition, low APB at the sites with MAP > 1500 mm might be caused by the rather low soil pH at these sites, as described in previous global analyses [17]. The pH ranged between 4.8 and 5.6 at these sites, whereas the mean across all sites was pH 6.0, and MAP and pH were strongly negatively correlated (Fig. 3). In contrast, it seems unlikely that the low APB observed at high MAP is due to anoxia fostered by high soil clay contents since clay had an overall positive effect on APB (Fig. 4B) and there was no significant interaction (P = 0.528) of clay and MAP in their relationships with APB (Table 2 and S1, Fig. 4B). In addition, high MAP did not lead to substantial leaching of nitrogen as we observed a positive correlation between MAP and soil total nitrogen (Fig. 3). The reason for this is that soil nitrogen is mostly covalently bound in soil organic matter [38] (see discussion below), in contrast to potassium and calcium. This leaves leaching of mobile nutrients from soil and low soil pH as the main reasons for low APB at high MAP.

In addition to MAP, we also found a significant unimodal model for APB as a function of AI (Table 1), indicating that APB is lower at the extreme arid and moist ends of the AI gradient compared to medium AI. This finding indicates that not only precipitation, but also the combination of precipitation and evaporation affects APB. At the moist end of the AI gradient, AI and APB are likely negatively related because AI affects soil water dynamics, and hence leaching processes, while at the arid end of the AI gradient, APB is likely limited by water availability. Our finding that the relationship between APB and AI follows a quadratic, unimodal relationship is in agreement with a recent meta-analysis [5].

## Interactions of abiotic variables

We found only partial support for our second hypothesis stating that MAP interacts with various abiotic variables in its relationship with APB. Against our second hypothesis, MAP and soil clay content did not significantly interact in their relationships with APB (Table 2 and S1, Fig. 4B). The reason for this might be that the sites with very high MAP (i.e., MAP > 1500 mm) did not have a soil clay content higher than 20%, and half of these sites had a clay content below 3.5%.

In accordance with the second hypothesis, MAP and MAT interacted significantly in their relationships with APB. Our analysis shows that different measures of temperature, such as MAT as well as the mean temperature of the driest and warmest quarter of the year were not significantly related to APB, neither through a linear nor through a unimodal function (Table 2). MAT was only significantly related to APB in interaction with MAP (Table 2 and Fig. 4). This indicates that temperature itself does not limit plant growth in global grasslands, and MAT is only significantly related to APB in interaction with MAP. MAT and MAP-squared ($MAP^2$) interacted significantly in a quadratic model (Table 2), showing that MAT modulates the relationship

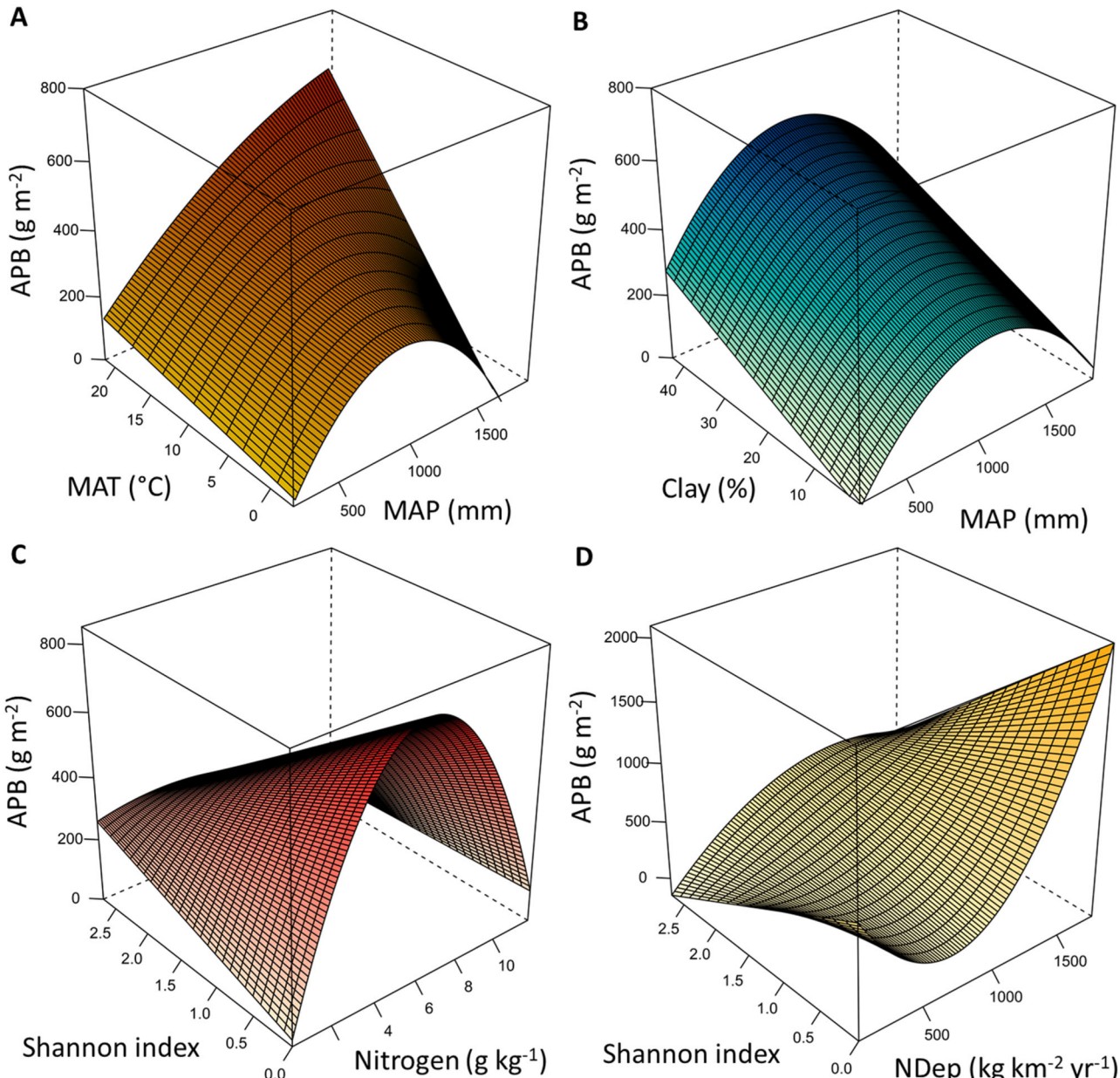

**Fig. 4 | Aboveground plant biomass (APB) as a function four pairs of predictors.** The plane presents APB that is predicted by a function containing one quadratic term. The models presented in panels **A**, **C**, and **D** contain a multiplicative interaction among the two predictors. The model in panel **B** does not contain an interaction. For model summaries see Table 2, and for model comparison see Supplementary Table S1. MAP: Mean annual precipitation, MAT: Mean annual temperature, Clay: soil clay content, Nitrogen: soil total nitrogen content, NDep: atmospheric nitrogen deposition.

between MAP and APB (Fig. 4A), as hypothesized. The model indicates that APB is decreased at sites with very high MAP compared to sites with medium MAP if MAT is low, but not if MAT is high (Fig. 4A). However, when acting in concert with other environmental factors, the relationship between APB and the interaction of MAP and MAT was not significant (Fig. 5), which is relevant for our fourth research question. It shows that single variables or interactions (of variables) that are significantly related with the dependent variable when considered in isolation might not be significantly related with the dependent variable in a more complex model. This is because the variability of the dependent variable can be captured to a large extent by other variables in models with multiple independent variables.

**Biomass as a function of nitrogen**
The relationship between APB and soil total nitrogen followed a quadratic function (Fig. 2B, Table 1), showing that beyond 5.96 g nitrogen kg$^{-1}$ soil,

APB was negatively related with nitrogen. Of the five sites with the highest soil total nitrogen content, one had medium MAP (943 mm), while the other four sites received high to very high MAP (1354, 1522, 1623, and 1877 mm). These five sites, located in the US, UK, and South Africa also had a high soil organic carbon content, suggesting that high MAP resulted in elevated soil organic matter (i.e., organic carbon and nitrogen) contents. This is supported by the fact that soil nitrogen and carbon stocks are positively correlated with MAP globally[39,40]. The nitrogen availability at the five sites with very high soil total nitrogen contents was likely not very different from the average nitrogen availability across all 55 sites, as indicated by the similar soil carbon-to-nitrogen ratios[41]. The soil carbon-to-nitrogen ratio at these sites ranged between 10.3 and 18.8 (median = 14.2), while the median soil carbon-to-nitrogen ratio across all 55 sites was 13.3. Thus, these findings indicate that low APB at the sites with very high soil nitrogen is not driven by nitrogen availability but mainly by the very high

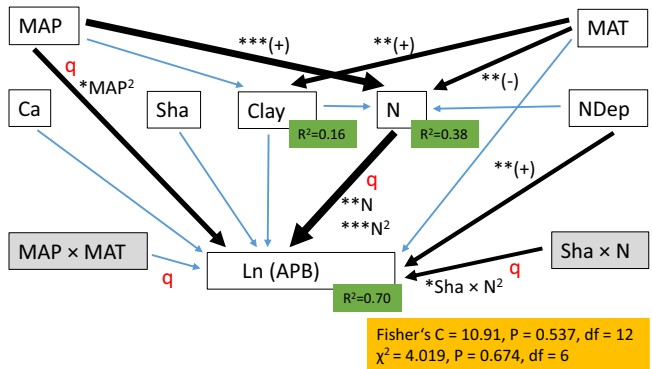

**Fig. 5 | Piecewise structural equation model (SEM) of log-transformed above-ground plant biomass (Ln(APB)).** The model includes quadratic functions (indicated by a red q) for the relationships between APB and nitrogen as well as APB and MAP. The model includes two multiplicative interactions of MAP and MAT as well as of the Shannon index and nitrogen in their relationships with APB, which are shown in gray boxes. Blue arrows indicate non-significant functions. Black arrows indicate significant functions. The thickness of the black arrows indicates the magnitude of the standardized path coefficients. Asterisks indicate the level of significance ($*P < 0.05$, $**P < 0.01$, $***P < 0.001$), and $(+)$ and $(−)$ indicate whether the slope of the linear functions is positive or negative. For the significant quadratic functions it is depicted which part of the function is significant, and the level of significance. The green boxes display the coefficient of determination ($R^2$) for the endogenous variables. The orange box displays the Fisher's C statistic, and the Chi-squared ($\chi 2$). The models were calculated based on centered data from the sites (centered site-level data), for all sites for which data on clay content is available ($N = 55$). MAP: Mean annual precipitation, MAT: Mean annual temperature, NDep: atmospheric nitrogen deposition, Clay: soil clay content, N: soil total nitrogen content, Sha: Shannon index of plant diversity, Ca: plant available soil calcium.

MAP at these sites, which causes low APB (as discussed in the previous section).

Soil nitrogen and MAP strongly covary (Fig. 3). Thus, to understand the relationship between APB and both MAP and soil nitrogen, and to address the fourth research question, we set up a SEM. The SEM shows that the direct effect of MAP on APB follows a quadratic function (see black asterisks and letters next to the quadratic relationship in Fig. 5), while MAP affects APB also indirectly through its linear effect on soil nitrogen (Fig. 5). Furthermore, the SEM indicates that APB is generally more strongly affected by soil nitrogen than by MAP (see thickness of the arrows depicting the path coefficients in Fig. 5). The reason for this is likely that soil nitrogen depends partly on MAP[39,40] and therefore captures a share of the variability of MAP, and additionally, nitrogen directly affects APB positively as a macronutrient. The latter is supported by a study on a subset of the grasslands studied here reporting that APB increased in response to nitrogen addition at most sites[6].

APB was not only significantly related to soil nitrogen but also to atmospheric nitrogen deposition (Tables 1 and 2, Fig. 4D). For the relationship between APB and atmospheric nitrogen deposition we found not only a quadratic but also a positive linear model (Table 1), indicating that atmospheric nitrogen deposition had an overall positive effect on APB. This is likely because atmospheric nitrogen deposition is beneficial for plant growth as a nitrogen source, although not at all levels of plant diversity (see below).

### Interactions of diversity and nitrogen
In accordance with our third hypothesis, the interaction of plant diversity and soil nitrogen was significantly related to APB, while plant diversity alone was not significantly related to APB (Table 2, Fig. 4, and Supplementary Table S1). The latter is in agreement with previous studies that explored a subset of the grassland sites investigated here and reported no significant linear relationship between plant diversity and APB[8,42]. Our results demonstrate that nitrogen modulates the relationship between APB and

plant diversity. APB was highest at low plant diversity and medium soil nitrogen content. Furthermore, APB was positively related with plant diversity at low levels of soil nitrogen, and negatively at medium levels of soil nitrogen (Fig. 4C). Similarly, APB increased strongly with atmospheric nitrogen deposition at low levels of plant diversity, and less so at very high levels of plant diversity (Fig. 4D). Our results are in accordance with several experimental studies showing that elevated nitrogen availability reduces the positive relationship between plant diversity and biomass in grasslands[31–33].

Several explanations for the observed relationships between plant diversity, nitrogen, and APB are not mutually exclusive. It could be that species with a high nitrogen-use efficiency are very productive at elevated levels of nitrogen and out-compete other species, leading to a high APB and low diversity[43–46]. It might also be that under low nitrogen availability, no plant species has enough nitrogen to outcompete others, leading to a high plant diversity at sites with low nitrogen availability and consequently low APB[47]. In addition, it could be that a very diverse plant community is beneficial for APB at low nitrogen availability since a high diversity of functional traits leads to complementarity in the use of resources (for instance, in the use of different chemical nitrogen forms)[30]. In general, our findings are in agreement with many studies about grassland ecosystems reporting a relationship between plant diversity and biomass in grassland[7,9,29,48], yet these studies did not explore the interaction between plant diversity and nitrogen. Our results show that nitrogen modulates the relationship between diversity and biomass, and that plant diversity and APB are negatively related at relatively elevated soil nitrogen contents. The findings indicate that the reason why several studies found no significant relationship between plant diversity and biomass[28] might be that they disregarded the interaction of plant diversity and soil nitrogen.

### Unimodal relationships and interactions
Taken together, we found some support for the first hypothesis since quadratic, unimodal functions could better describe the relationships between APB and MAP, APB and AI, APB and elevation, as well as APB and soil nitrogen than linear, monotonic functions (Table 1). Our results suggest that APB has an optimum at the medium range of these variables. According to Shelford's theory (or law) of tolerance[49], the relationship between biomass of a species and an abiotic ecosystem property follows a unimodal function since biomass decreases with the extent to which the abiotic conditions diverge from the optimum condition of the species. Shelford formulated the law of tolerance[49] for species, and it might seem questionable whether this theory holds true for communities because of differences in optimum conditions among species. However, our results indicate that plant community biomass decreases towards the outer ranges of some environmental variables, such as MAP, AI, elevation, and soil nitrogen. The reason for this seems to be that very high MAP and very humid conditions have negative effects on most plant species in grasslands.

Against our first hypothesis, for some variables we found no unimodal relationship with APB. For example, we found no quadratic, unimodal function describing the relationship between soil APB and pH (Table 1), indicating that the outer pH ranges (pH 4.0 and 8.3) were not extreme enough to cause decreased APB[50]. Furthermore, for APB as a function of clay, we found both a significant linear and a quadratic model, indicating that clay had mainly a positive effect on APB (Table 1, Fig. 4B). This is likely because clay is the most important source of all nutrients, except for nitrogen, as indicated also by the positive correlations of clay content and plant-available soil potassium and calcium (Fig. 3) as well as the positive relationship between plant-available calcium and APB (Table 1). Soil clay content was not significantly correlated with MAP, but it was positively correlated with MAT (Figs. 5 and 3), suggesting a positive influence of temperature on the formation of clay-sized minerals, as described in previous studies[51].

Our study demonstrates important interactions between abiotic and biotic variables in global grasslands. Specifically, our results demonstrate that plant species diversity and nitrogen interact significantly in their relationships with APB (Table 2 and Fig. 4). Furthermore, the study shows that

APB is not significantly related with MAT (Table 1), but with the interaction of MAT and MAP (Table 2). Our results highlight the importance to account for the interactive and unimodal relationships between plant biomass and several environmental variables when analyzing and modeling plant biomass at the global scale. Including interactive and unimodal relationships in global vegetation and carbon models likely improves their ability to predict primary production at the global scale.

## Conclusions

We found that APB was related through unimodal functions with several environmental factors that interacted with other ecosystem characteristics in their relationships with APB. Specifically, we observed that APB was related to MAP through a quadratic function with a maximum at 1138 mm. This is likely because very high MAP has adverse effects on APB since it causes a low soil pH and leaching of mobile nutrients, such as potassium, from soil. Furthermore, APB was related with soil nitrogen through a unimodal function, and nitrogen and plant diversity interacted in their relationships with APB. Plant diversity and biomass were positively related at low levels of nitrogen and negatively at medium levels of soil nitrogen. Our study has important implications as it suggests that models assuming a linear positive relationship between plant biomass and MAP at the global scale will likely overestimate biomass in grasslands with high precipitation. Furthermore, our results show that models of APB should include an interaction of plant diversity and nitrogen since nitrogen can reverse the relationship between plant biomass and plant diversity. This is important for the inclusion of plant biomass in global vegetation and carbon models.

## Methods

### Study sites

All 116 grassland sites studied here are natural or semi-natural grasslands that are part of the Nutrient Network Global Research Cooperative[52] (https://nutnet.org). MAP at the sites varies between 192 and 1877 mm, while MAT varies between −6.6 and 27.3 °C, soil clay content varies between 0.8 and 44.6%, and soil pH varies between pH 4.0 and 8.3. For this study we chose data that were collected the year before any experimental treatment started, meaning sites were not experimentally manipulated at the time of data collection.

### Sampling, measurements, and climate data

All sites followed the same sampling protocol, and the data were collected between 2007 and 2020. At each site, on average about 30 plots were established (10 to 60 plots per site) that have a size of $5 \times 5$ m. In total, there were 3380 plots distributed across 116 sites.

Plant species composition was determined in a randomly designated $1 \times 1$ m subplot within each $5 \times 5$ m plot at peak biomass. In the same $1 \times 1$ m subplot, cover was estimated visually to the nearest 1% for every species overhanging the subplot.

Live vascular plant aboveground biomass (hereafter aboveground plant biomass or APB) was measured at peak biomass (i.e., at the specific time of the year when aboveground plant biomass is highest). This was done destructively by clipping all aboveground biomass at ground level of plants rooted within two $1 \times 0.1$ m strips (for a total of 0.2 m²) adjacent to the $1 \times 1$ m subplot where plant species composition was determined. All biomass was dried at 60 °C to constant mass before weighing to the nearest 0.01 g. Data on aboveground plant biomass and plant species composition were collected at all 116 sites.

Soil samples were collected in the $5 \times 5$ m plots by taking three soil cores (2.5 cm diameter) at a depth of 0–10 cm. The three cores were pooled to make one sample per plot. Root fragments were removed, and the soils were air-dried and sieved ( < 2.0 mm) prior to any analysis. The samples were analyzed for total organic carbon and total nitrogen using an elemental analyzer (Costech ECS 4010 CHNSO Analyzer). Plant-available soil phosphorus (P), potassium (K), and calcium (Ca) were extracted from soil according to the Mehlich-3 protocol[53] and quantified using Inductively Coupled Plasma Mass Spectrometry. Soil pH was measured in a 1:1 soil:

water (v/v) suspension. Soil texture, i.e., clay, silt and sand, was measured using the Bouyoucos method. All soil samples were analyzed in the same laboratory (Waypoint Analytical Laboratory, Memphis, Tennessee, USA). Data on soil chemistry and clay content were collected for 55 of the 116 sites.

We obtained data on precipitation and temperature from Worldclim 2.0[54]. Specifically, we extracted data on mean annual precipitation (MAP), mean annual temperature (MAT), mean precipitation of the wettest quarter of the year (PWet), mean temperature of the driest quarter of the year (TDry), mean temperature of warmest quarter of the year (TWarm), and maximum temperature of warmest month (TMax). Data on potential evapotranspiration (PET) was obtained from the Consultative Group for International Agricultural Research (CGIAR), and data on atmospheric nitrogen deposition from Ackerman et al.[55] for all sites. We calculate the aridity index (AI) by dividing MAP by PET.

### Calculations and data analyses

We calculated two indices of plant species diversity, the Shannon-Wiener diversity index (called Shannon index hereafter) and the Simpson index as well as species richness and evenness from the plant species composition data collected at the plot scale, using the R package vegan (version 2.6-4)[56]. We choose the Shannon and the Simpson indices as two contrasting measures of diversity. The first is more sensitive to differences in rare species abundance, while the latter is more sensitive to differences in the most abundant species[57].

We calculated arithmetic means of plant biomass, plant diversity, species richness and evenness as well as soil properties across all plots of each site (called site-level data or site means in the following). This is common practice in global studies in ecology[8]. We aggregated the data at the site-level because the climate variables vary among the sites but not among the plots of one site. In addition, different plots of one site are not independent of each other (which is a pre-requisite for regression analysis). The aim of our study is to understand how plant biomass is related to abiotic variables and plant diversity in grasslands spanning a wide range of climate conditions. Thus, the scale of inference of our study is the global scale, and not the local scale of a single site. Our approach is based on the common understanding that variability at different spatial scales is caused by different drivers[58], which implies that the analysis of small-scale variability has very limited value for understanding variability occurring at the large scale.

A correlation matrix was calculated and visualized using the R package corrplot (version 0.92). All variables were mean centered before models were calculated to avoid covariance of the quadratic and non-quadratic term in the quadratic models[59]. In addition, we created scatterplots for regressions that show the fitted values of the model vs. the residuals of those fitted values to evaluate heteroscedasticity. APB was transformed by calculating its natural logarithm. This was done because the residuals of the regression models were not normally distributed prior to the log-transformation and because of heteroscedasticity. We calculated the following models of APB and compared them based on the Akaike Information Criterion (AIC), and considered models with a smaller AIC ($\Delta$AIC < 2) to fit the data better[60].

Linear model with one predictor: Y ~ X

$$Y = \beta_0 + \beta_1 X + \varepsilon \qquad (1)$$

Model with one predictor and one quadratic term: Y ~ X + X²

$$Y = \beta_0 + \beta_1 X + \beta_2 X^2 + \varepsilon \qquad (2)$$

Subsequently, models with two predictors and one or two quadratic terms were calculated, according to our hypotheses and the results of the previous regression analyses. We calculated these models with and without multiplicative interaction of the two predictors. This was done only for pairs of predictors that were not significantly ($P > 0.05$) correlated[61]. We fitted the following models (shown below in the shorter R notation and in mathematical notation) to our data and compared the fit based on the AIC.

Model with one quadratic term and no interaction: $Y \sim X_1 + X_1^2 + X_2$

$$Y = \beta_0 + \beta_1 X_1 + \beta_2 X_2 + \beta_3 X_1^2 + \varepsilon \qquad (3)$$

Model with one quadratic term and multiplicative interactions: $Y \sim X_1*X_2 + X_1^2*X_2$

$$Y = \beta_0 + \beta_1 X_1 + \beta_2 X_2 + \beta_3 X_1^2 + \beta_4 X_1 X_2 + \beta_5 X_1^2 X_2 + \varepsilon \qquad (4)$$

Model with two quadratic terms and no interaction: $Y \sim X_1 + X_1^2 + X_2 + X_2^2$

$$Y = \beta_0 + \beta_1 X_1 + \beta_2 X_2 + \beta_3 X_1^2 + \beta_4 X_2^2 + \varepsilon \qquad (5)$$

Model with two quadratic terms and multiplicative interactions: $Y \sim X_1*X_2 + X_1^2*X_2 + X_1*X_2^2$

$$Y = \beta_0 + \beta_1 X_1 + \beta_2 X_2 + \beta_3 X_1^2 + \beta_4 X_2^2 + \beta_5 X_1 X_2 + \beta_6 X_1^2 X_2 + \beta_7 X_1 X_2^2 + \varepsilon \qquad (6)$$

For models for which we found a significant interaction based on the log-transformed APB data, we calculated the same model also with the non-transformed data, in order to exclude that the interaction is a result of the log-transformation[61]. We visualized the best models with two predictors in 3-dimensional plots using the R package lattice (version 0.22-5) based on the non-transformed site-level data.

## Piecewise structural equation modeling

In order to better understand how different ecosystem properties act in concert and to evaluate direct and indirect effects of MAP, we conducted piecewise structural equation (SEM) modeling, using the R package piecewiseSEM (version 2.3.0)[62]. We choose piecewise SEM because it allows us to include interactions among variables (in contrast to other path modeling approaches). All variables were mean centered before the models were fitted[60]. In addition, APB was transformed by calculating its natural logarithm. This was done because the residuals of the regression models were not normally distributed prior to the log-transformation. All SEMs were fitted to the site level-data for all complete sites (i.e., sites for which data on clay content was available). We evaluated the fit of different versions of the model to our data using the Akaike Information Criterion (AIC).

According to our hypotheses and the results of the previous regression analyses, we set up a piecewise SEM of APB as a function of MAT, MAP, the Shannon index, soil clay and nitrogen contents as well as atmospheric nitrogen deposition. We included MAT and MAP, but not AI in this model because AI is calculated based on MAP and evapotranspiration, and the latter is related to MAT. We calculated this model with linear and quadratic functions for the relationships of MAP and APB as well as soil nitrogen and APB, and we selected the model version with the lowest AIC. We included interactions of MAT and MAP as well as Shannon index and soil nitrogen in the SEM since we had observed interactions of these variables in the previous regression analyses. We further optimized the model by including either plant available soil P, K, or Ca, or all three nutrients as additional predictors of APB, and we selected the best-fitting version of these four models based on the lowest AIC. In addition, we tested a version of this model in which soil nitrogen and clay are not only affected by MAP and MAT (without interaction) but also by the interaction of the two variables. However, this model had a higher AIC ($\Delta$AIC > 2) than the model in which soil nitrogen and clay are affected by MAP and MAT without interaction. All data analyses were conducted in R (version 4.2.1)[63].

## Statistics and Reproducibility

The 116 grassland sites studied here are located on six continents. At all sites, data were collected according to the same protocol[52] in 10–60 plots of a size of 5 × 5 m. In total, there were 3380 plots distributed across 116 sites. We calculated arithmetic means of abiotic and biotic variables across all plots of each site, and all statistical analyses were conducted based on these means. We calculated regression models in R (version 4.2.1)[63], and we evaluated the fit of different models to our data using the AIC. All data and the R code to reproduce the results are publically available[64].

## Reporting summary

Further information on research design is available in the Nature Portfolio Reporting Summary linked to this article.

## Data availability

All data are available at this repository. https://doi.org/10.5281/zenodo.14509903.

## Code availability

All R code for reproducing the results is available at this repository. https://doi.org/10.5281/zenodo.14509903.

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

## Acknowledgements

This work was generated using data from the Nutrient Network experiment, funded at the site-scale by individual researchers. Coordination and data

management have been supported by funding to E.T.B. and E.W.S. from the National Science Foundation Research Coordination Network (NSF-DEB-1042132) and Long Term Ecological Research (NSF-DEB-1234162 to Cedar Creek LTER) programs, and the Institute on the Environment (DG-0001-13). We also thank the Minnesota Supercomputer Institute for hosting project data and the Institute on the Environment for hosting Network meetings. Soil analyses were supported by funds from Oregon State University and University of Minnesota to E.T.B. and E.W.S. and by USDA-ARS grant 58-3098-7-007 to ETB. M.S. acknowledges funding from the European Research Council (ERC) (grant number 101043387). J.A.C. acknowledges funding from the ERC under the European Union's Horizon 2020 research and innovation program (grant agreement No. [101002987]). N.E. acknowledges the support of iDiv, which is funded by the German Research Foundation (DFG – FZT 118, 202548816), as well as by the DFG (Ei 862/29-1; Ei 862/31-1). A.E. acknowledges the Finnish Academy (project no 351089). Y.L. acknowledges MPG Ranch for funding establishment, maintenance and data collection of NutNet sites. L.L. acknowledges the Estonian Academy of Sciences (research professorship for Arctic studies) and Adam Mickiewicz University Polar Station (Petuniabukta). K.J.K. acknowledges Konza Prairie LTER funding from NSF 2025849, NSF 1440484, NSF 0823341, NSF 0218210. M.P. acknowledges the Estonian Research Council (PRG609) and the Centre of Excellence AgroCropFuture. N.G.S. acknowledges support from Texas Tech University and the US National Science Foundation (DEB-2045968).

## Author contributions

M.S. developed and framed research questions, analyzed data, and wrote the paper. E.T.B. and E.W.S are Nutrient Network Coordinators and site level coordinators, and contributed to paper writing and editing. S.B., J.D.B., C.C., J.A.C., C.R.D., N.E., A.E., N.H., Y.H., S.E.K., K.J.K., L.L., Y.L., J.P.M., H.M., M.P., P.L.P., A.C.R., N.G.S., C.S., G.F.V., R.V., L.Y. A.L.Y., H.Y. are site level coordinators and contributed to paper editing and writing (see Supplementary Table S3).

## Funding

## Competing interests

The authors declare no competing interests.

## Additional information

[1]Dept of Soil and Environment, Swedish University of Agricultural Sciences (SLU), Uppsala, Sweden. [2]Centre for Ecological Sciences, Indian Institute of Science, Bangalore, India. [3]School of Environmental and Forest Sciences, University of Washington, Seattle, USA. [4]Dept of Ecology, Evolution, and Behavior, University of Minnesota, St Paul, USA. [5]Scientific Services, Ezemvelo KZN Wildlife, Cascades, South Africa. [6]School of Life Sciences, University of KwaZulu-Natal, Scottsville, South Africa. [7]Dept of Geography, King's College London, London, UK. [8]School of Agriculture, Food & Ecosystem Sciences, University of Melbourne, Parkville, Australia. [9]Fenner School of Environment & Society, The Australian National University, Canberra, Australia. [10]School of Life and Environmental Sciences, The University of Sydney, Sydney, Australia. [11]German Centre for Integrative Biodiversity Research (iDiv) Halle-Jena-Leipzig, Leipzig, Germany. [12]Leipzig University, Institute of Biology, Leipzig, Germany. [13]Ecology & Genetics, University of Oulu, Oulu, Finland. [14]Mammal Research Institute, Dept of Zoology & Entomology, University of Pretoria, Pretoria, South Africa. [15]Ecology and Biodiversity Group, Dept of Biology, Utrecht University, Utrecht, The Netherlands. [16]Dept of Biology, University of North Carolina Greensboro, Greensboro, USA. [17]Dept of Biodiversity and Nature Tourism, Estonian University of Life Sciences, Tartu, Estonia. [18]MPG Ranch and University of Montana, Montana, USA. [19]Dept of Biology, Texas State University, San Marcos, USA. [20]Dept of Biology, McDaniel College, Westminster, USA. [21]Institute of Ecology and Earth Sciences, University of Tartu, J. Liivi 2, Tartu, Estonia. [22]National Institute of Agricultural Technology (INTA), Rio Gallegos, Argentina. [23]Swiss Federal Institute for Forest, Snow and Landscape Research WSL, Birmensdorf, Switzerland. [24]Dept of Biological Sciences, Texas Tech University, Lubbock, USA. [25]Lancaster Environment Centre, Lancaster University, Lancaster, UK. [26]Dept of Terrestrial Ecology, Netherlands Institute of Ecology, Wageningen, the Netherlands. [27]Instituto de Investigaciones Fisiológicas y Ecológicas Vinculadas a la Agricultura (IFEVA), CONICET, Faculty of Agronomy, University of Buenos Aires, Buenos Aires, Argentina. [28]Dept Ecology, Evolution, and Marine Biology, University of California Santa Barbara, Santa Barbara, USA. ✉e-mail: marie.spohn@slu.se

