## [Transparent Peer Review file · Communications Biology]

Interactive and unimodal relationships between plant biomass, abiotic factors, and plant diversity in global grasslands

Corresponding Author: Professor Marie Spohn

Version 0:

Reviewer comments:

Reviewer #1

(Remarks to the Author)

Dear Editor,

I have carefully revised the manuscript titled "Unimodal relationships between plant biomass and the environment in global grasslands".

Here, the authors analysed a global grassland database and tested unimodal and interactive relationships between aboveground biomass and some of its potential drivers.

In overall, I think the authors have addressed a relevant topic in ecology, and the conclusions of the study are important for the field and for a broader audience (e.g., modeler of global vegetation and carbon). Maybe the authors should highlight this a little bit more. In my personal view, I agree with the authors, curves and interactions are often neglected in ecology, so we must insist they are important, and provide evidence.

However, I see some problems, important ones in my opinion, so I would suggest a deep revision. These problems are related with the concrete questions of the study and the statistical analyses chosen. I will try to explain how, in my view, this is a major problem as it affects to the global coherence of the paper.

First, and most important, the authors carried out a piecewise SEM model, with some interesting results. However, it does not fit with any of the concrete questions stated at the end of the introduction. Then, we don't know which hypothesis were tested by the authors with this analysis.

Second, the authors argue in the methods that they "aggregated the data at the site level because the climate variables vary among the sites but not among the plots of one site. In addition, different plots of one site are not independent of each other (which is a pre-requisite for regression analysis), which is why we conducted regression analyses based on the plot-level data" (L 434-438).

I disagree with their decision, as there are other well-known specific options to deal with this type of data, like mixed linear models (piecewise SEM models and other SEM options allow to include random effects too). I think authors should, at least, to include a convincing justification of why they chose this procedure and not mixed linear effects models. I want to highlight that, by aggregating the data at the site level, the sample size is reduced, so is the power of the analysis. Note that in the piecewise SEM model, some variables like soil K or P were discarded for having only marginal, but not significant effects. I think this is a problem because in the discussion the authors posit these variables could actually have an effect on aboveground biomass (L 252).

In summary, I would say the manuscript in general needs to improve the link of the introduction and discussion sections with the statistical analyses and their results. It is quite consistent but sometimes there are important gaps. These can be filled by both improving the introduction and the discussion, as well as the statistical approach and/or its justification.

I also have some doubts about the choice of the Shannon's index as a proxy of plant biodiversity, that I have explained in the detailed comments.

Finally, I would suggest the authors to test more climate and plant diversity variables and to change the title to something less descriptive and more eye-catching.

In the lines bellow, I include more detailed comments. I hope they are useful to improve the manuscript:

Title:

You should change the title since not everybody who looks for curve relationships is going to look for the word "unimodal". also, you have found much more than that. You should look for a more eye-catching title.

Introduction:

In general, its quite well exposed, but I think it's sometimes incomplete (you don't mention MAT until the line 115, and quite briefly if you compare with other variables). I am not saying you should dedicate the same text to all variables, but in general I find the explanation about some variables is somehow scarce.

L 83: Here I would suggest to remark this is a common problem with interactions (an unimodal relationships) in ecology "widely assumed, but rarely tested" (Mantyka-pringle et al. 2012). That's why this paper is important! Also, I would highlight the connection with vegetation and soil models as you do in the conclusions.

See the following references:

Mantyka-pringle, C. S., Martin, T. G., & Rhodes, J. R. (2012). Interactions between climate and habitat loss effects on biodiversity: A systematic review and meta-analysis. *Global Change Biology*, 18(4), 1239–1252. <https://doi.org/10.1111/j.1365-2486.2011.02593.x>

Rillig, Matthias C., Masahiro Ryo, Anika Lehmann, Carlos A. Aguilar-Trigueros, Sabine Buchert, Anja Wulf, Aiko Iwasaki, Julien Roy, and Gaowen Yang (2019), The role of multiple global change factors in driving soil functions and microbial biodiversity." *Science*, 366, 886{890, doi: 10.1126/science.aay2832.

Spiegelberger, Thomas, Francois Gillet, Bernard Amiaud, Aurélie Thiebault, Pierre Mariotte, and Alexandre Buttler (2012), How do plant community ecologists consider the complementarity of observational, experimental and theoretical modelling approaches?" *Plant Ecology and Evolution*, 145, 4 doi:10.5091/plecevo.2012.699.

Sirami, Clelia, Paul Caplat, Simon Popy, Alex Clamens, Raphael Arlettaz, Friedrerich Jiguet, Lluís Brotons, and Jean Louis Martin (2017), Impacts of global change on species distributions: obstacles and solutions to integrate climate and land.

L 95: This is quite good hypothesis and is quite well exposed.

L 102 In this case, I would expect some interactions with precipitation.

L 108: Incomplete if we compare to the previous variables (see pH). I think this is important, since for pH is quite well-known but for nitrogen is not so clear. I think you develop something about soil N below. Maybe you should mention it here instead.

L 114 Then we don't expect unimodal relationships (or not only) but interactions. Maybe that should be mentioned at the beginning of the paragraph. Actually, an unimodal relationship is an interaction of an explanatory variable with itself.

L 112 Good hypothesis again.

L 125: Here it is, the soil N. Anyway, I miss some explanations about why the aboveground biomass should change with soil N. By the way, the authors focus only on total N (which is mostly organic N in natural grasslands).

When I read the first mention to soil N in line 108, I wrote: "I will wait also some interaction with plant diversity, as the effects of temporal and spatial niche complementarity of species (e.g. grasses and legumes, but also forbs) in nutrient acquisition are widely (but not completely) studied". Maybe some phrases similar to this one could help the readers.

L 130 This is also known, interactions are widely assumed, but rarely tested, as I mentioned above. This is why I would suggest the authors to include a table with some recent or classical works addressing the main drivers of aboveground plant biomass at wide (from regional to global) scales with climate ranges, a brief list of the variables tested, and if they tested interactions or not. I know this implies some work, but also would be a powerful way to transmit the reader the importance of this study.

L 137 I would definitely include temperature seasonality (maybe the relationship is unimodal too) and precipitation on the warmest quarter (i.e. mean summer precipitation) at least. See my comments to the methods.

L 141 Did you try with climate variables?

L 143 Then you performed a test/model that does not answer any of your questions (the piecewise SEM). I assume is because you are interested in provide evidence for causal models, you want to show direct and indirect effects of some variables (MAP), and of course, to put all the variables together (before you tested only individual correlations). This is important because is the only way to justify your method based of pairwise correlations instead of working directly with a single linear model that includes all predictors as candidate explanatory variables. This last approach could discard any variable (E.G MAP) that could affect AB biomass through other variables.

L 144 Were this data used before for other papers? if yes, it is ok, but mention then.

Results:

I think they are well exposed.

L 153: You can say the quadratic model performed better ($\Delta AIC > 2$) and that would be enough (Burnham and Anderson 2002).

L 163 I think it is a good idea to write the number of samples ($n = 116$ or 55 , together with the P and the R²). So that is easy for the reader as they don't have to remember which analyses were done with each set.

L 173 Table 1, you miss a space between Table and 1.

L 193 This seems unnecessary to me. Actually, you transformed the variable to avoid error I or II types, that can come from deviation of the normality of your variable/residuals.

L 202 I think this suggest MATxMAP does not affect APM directly, but through other variables. This is what you test in a SEM. I think you should, at least, explain if you can check this or not.

Discussion:

As I have already mentioned, I would say the manuscript in general needs to improve the link of the introduction and discussion sections with statistical analyses and the results. Here I provide some comments/suggestions with the hope they help the authors to improve the discussion.

229 This is a common problem in ecological papers. See

Maire, V., Wright, I. J., Prentice, I. C., Batjes, N. H., Bhaskar, R., van Bodegom, P. M., Cornwell, W. K., Ellsworth, D., Niinemets, Ü., Ordonez, A., Reich, P. B., & Santiago, L. S. (2015). Global effects of soil and climate on leaf photosynthetic traits and rates. *Global Ecology and Biogeography*, 24(6), 706–717. <https://doi.org/10.1111/geb.12296>

L 239 I think this could be clearer in the writing. Also, you specify the temperature but not the precipitation limits.

L 245 I think you want to write "Consequently, not "furthermore."

L 260 You only talk about nitrogen below. The point on potassium is interesting, because as it is not part of organic structures its cycle has differences with N and other nutrients.

McGrath, Joshua, John Spargo, and Chad Penn (2014), *Soil Fertility and Plant Nutrition*, 166{184. Elsevier, doi: 10.1016/B978-0-444-52512-3.00249-7.

Sardans, Jordi and Josep Penuelas (2015), Potassium: A neglected nutrient in global change. doi: 10.1111/geb.12259.

I think you could reinforce your interpretation by testing K relationships too, and include it in the piecewise model. Otherwise, you should mention why not. I understand potassium was discarded in the piecewise SEM model. This is why I think mixed models are more appropriate for your dataset than site aggregation.

L 261 And you did not find interaction between MAP and Clay, did you? I would mention that, because that as it reinforces your argument. Interactions between MAP and Clay are common in literature, by the way:

Rodríguez, A., Canals, R. M., Plaixats, J., Albanell, E., Debouk, H., Garcia-Pausas, J., San Emeterio, L., Ribas, À., Jimenez, J. J., & Sebastià, M.-T. T. (2020). Interactions between biogeochemical and management factors explain soil organic carbon in Pyrenean grasslands. *Biogeosciences*, 17(23), 6033–6050. <https://doi.org/10.5194/bg-17-6033-2020> and references therein.

L 283 Is there no way to represent the MAT*MAP interaction in your piecewise model? Maybe checking the relative importance of MAP and MAT and its interaction (e.g. relaimpo package) could give us some light. Or to include the interaction but not affecting directly to plant biomass.

L 302 Is it possible to include a SEM for the 116 sites, only for the variables that are available? at least to put in the supp mats and compare.

306 Logical, soil variables will affect the vegetation variables (including biomass) more than climate variables, especially if these are taken from a map (Rodriguez et al. 2022 found the same for plant trait indices)

Rodríguez, A., de Vries, F. T., Manning, P., Sebastià, M. T., & Bardgett, R. D. (2022). Soil Abiotic Properties Shape Plant Functional Diversity Across Temperate Grassland Plant Communities. *Ecosystems*. <https://doi.org/10.1007/s10021-022-00812-2>

L 311 Could you provide some explanation? For instance, because soil total nitrogen is not a perfect indicator of nitrogen availability. Most N is immobilised and then the available N probably does not satisfy the whole plant nitrogen demand. But maybe you can find a better one.

L 319. What about Richness, Simpson, evenness? See comments in methods section.

L 334 This is quite important, the dominance effect primes in high nutrient abundance scenarios, and the niche complementarity in the low nutrient scenarios. Then I suggest you to discuss if this is consistent with studies about functional diversity and plant biomass, because most functional diversity indices are actually hypotheses of one of these effects (e.g. CWMs for dominance effects and Rao's Q for niche complementarity).

L 339: Did these authors check for the interaction, or only the main effect? If your results are a novelty, you should highlight it.

L 370 This is very accurate. Sometimes this paper has very high quality and interesting details.

L 344 or other nutrients.

Conclusion

Maybe the authors want to change something after the revision.

Methods:

I think they are very well explained, though as I mentioned, I disagree with the site aggregation and with the way of choosing a plant diversity index.

L 421 Were is pH?

L 423 From Worldclim or Worldclim 2.0 (which is the same approach, but the second was built with much more sites, and is the one you must use). This database has 18 climatic variables, so there are some possibilities to test other climate variables. I would encourage you to test temperature or precipitation seasonality, for instance, as these are the same case as interactions or unimodal effects, they are not tested very often.

L 429 Why Shannon's diversity is used? I miss some justification.

I assume it is because the authors want to summarise richness and evenness together (number of species and their proportions). However, using both richness and the Camargo's or Pielou's evenness (you can choose the one that less correlates with richness) instead of just one index would be more informative than using a composite index.

They could even test which approach is best for describing the relationship between plant diversity and aboveground productivity: richness and evenness, Shannon or Simpson. Anyway, sometimes Shannon index is highly correlated with species richness, that's why I think they should justify this.

Note that you calculate Shannon's index with a package that explains Simpson's index could be preferable: typing "? diversity" in R: "Better stories can be told about Simpson's index than about Shannon's index, and still grander narratives about rarefaction (Hurlbert 1971). However, these indices are all very closely related (Hill 1973), and there is no reason to despise one more than others... In particular, the exponent of the Shannon index is linearly related to inverse Simpson (Hill 1973) although the former may be more sensitive to rare species).

For justifying your decision, you can check also

Roswell, M., Dushoff, J., & Winfree, R. (2021). A conceptual guide to measuring species diversity. *Oikos*, 130(3), 321-338.

L 444 you tested the normality of residuals, but you also should provide some evidence of their homoscedasticity, as heteroscedasticity could induce type I error in your models. Also, some information about the influence of outliers in the model estimates should be advisable (e.g. a plot of Cook's distance and Leverage).

L 447 This is unclear. I guess you mean you fit the equations in this order (from less complex to more complex) and you accept the more complex equation if it improves the previous one when the AIC difference is higher than two.

L 547 In my view, the threshold should be based on R square too (or in variance inflation factor, see the HH package). In you base on p value, maybe you are discarding some interactions just because the correlation is significant but weak (e.g. R squ = 0.1 or 0.2 can give a p-value < 0.05 and I think you can still test the interaction with no problem).

L 461 These equations are misleading as they are slightly different to the true equations detailed one line below. This could confound some readers.

L 471, In the introduction you should explain what do you aim with this analysis.

L 481 In supplementary material I suggest you to include some diagrams (maybe just with the diagrams and the AIC, or a table of AICs in addition) to give an idea of how was your model selection.

L 492 From my point of view, this is a problem, because you explain things about available potassium in the discussion, without mentioning that it is not significant and out of your final model, and you did not mention this.

Tables

Table 1: maybe you should move the information for all sites analyses to supplementary material, to say your results were consistent, and focus on the 55 sites.

Where is potassium?

Table 2: I would write delta AIC respect to the model of reference in the previous table. Otherwise, AIC itself is meaningless.

Figure 2: I know is in the table, but I suggest to mention the R2 and the p-value in the figure or in the caption.

Figure 5: the mention of AIC in the box is meaningless, unless you have a table of different SEM models with different AICs

Reviewer #2

(Remarks to the Author)

This study explored the relationship between grassland production and environmental interactions using data from 116 grasslands on six continents. The study found that plant biomass has unimodal relationships with factors like mean annual precipitation (MAP) and soil nitrogen. Plant diversity and biomass are positively related at low nitrogen levels and negatively at high levels.

The data collection was done between 2007 and 2020, it needed to be clearly mentioned which month or mowing frequency/grazing condition or what stage of the grass growth.

The authors discussed the impact of abiotic factors on grass biomass. However, the reasons for avoiding elevation and slope should have been mentioned or tried to be used in the modelling process.

Further land use type (e.g., agriculture, nature protected, grazing) and land use intensity (intensively or extensively managed, fertilised or not) could change plant biomass available. Yet needed to be more information could not be found in the manuscript about those factors.

Consideration should be given to the potential of growing degree days as a more effective method for establishing the relationship between biomass and temperature, as opposed to MAT.

Version 1:

Reviewer comments:

Reviewer #1

(Remarks to the Author)

Dear Editor,

I have carefully revised the manuscript and the authors answer to mi revision. In general, they answered satisfactorily to my comments. I think the changes they made have improved the manuscript, especially the discussion section, which seems now quite convincing and transmit the ideas in such an efficient way.

I am not totally convinced about the title, that now seems somehow vague. I would suggest something more explicit, like "The importance of non-linear and interactive relationships between plant diversity and abiotic factors for plant biomass in global grasslands". Maybe the editor or the authors prefer the current title or want to suggest another one.

I also think that the last research question they added is incomplete, as they use a structural equation model, then "to which extent do different variables (that are related with APB through quadratic functions) explain variation in APB when considered together?" is not only what they test. They could have used a multiple linear model for this purpose. However, their election of SEM is accurate because with a conventional linear model they would take the risk of discarding some variables that are related or have causal relationships between them. A good example in this case is MAT and Clay. I agree this is implicit in the manuscript but should be made explicit in the different sections (introduction, methods, in the discussion, why not...) at least briefly. You could take a look at the "Hierarchy of controls over function" hypothesis exposed here:

De Vries, F. T., Manning, P., Tallwin, J. R., Mortimer, S. R., Pilgrim, E. S., Harrison, K. A., ... & Bardgett, R. D. (2012). Abiotic drivers and plant traits explain landscape-scale patterns in soil microbial communities. *Ecology letters*, 15(11), 1230-1239.

Maybe the ideas there can help.

My apologies if I was not clear enough about these two last points in my previous revision. Anyway, I insist in the great improvement of the manuscript.

Additionally, after the changes you have done, I believe that the end of the discussion, you could take profit on your SEM to finish with some comments of the global picture you found in your paper, and that this is highly relevant for improving vegetation and, why not, carbon models too. A short paragraph with a few references should be enough. You could even use this idea to improve your last question at the introduction, as in the rest of your paper you test a number of ABP drivers one by one (or in paris, in some cases), but the idea of the paper is to propose improvements in biomass and carbon models, and these consider multiple variables as you do with your SEM.

Concerning the SEM, something maybe I would mention more explicitly is that the indirect effects of MAP seemed to be linear, but the direct effects are unimodal.

I would also like to suggest some minor changes, in case the authors want to consider them.

L. 116. I wrote that in my previous revision, but maybe it's more scientific to say "have multiplicative effects with itself". I usually use the phrase "in unimodal relationships, in which one explanatory variable interacts with itself" to try to help people to understand, but inter-action implies minimum 2 actors or components.

L 124. Maybe a ref. here...

L 151 I have already exposed my view about this question above.

L163 Also, see above. This is not the only thing you are doing with a SEM, and in fact, it is well done, so a better explanation would put into value the work done.

L 242 "APB and MAP followed a quadratic model." Maybe, they follow a quadratic relationship? Not a model.

L 426: To be honest, I did not understand if in Fig. 4B Clay is represented linearly or in an unimodal relationship. Checking table 1, I see the quadratic model its significant, but the AIC is higher than the linear model. Maybe it would be clearer to say the quadratic relationship of clay is significant but negligible in terms of size, then the relation is mostly linearly positive, as you expose.

L. 430-433 the second reason for using SEM...

Table 1: I don't see the relevance of showing adjusted R2 here, since this is a correction for a high number of variables. They are very similar because you only have 1 or 2 terms in the linear or quadratic model, respectively.

Also, maybe you want to simplify this table by suppressing AIC in the linear model table and replacing AIC by Delta AIC in the quadratic model, being $\Delta AIC = AIC_{\text{quadratic model}} - AIC_{\text{linear model}}$, and showing negative values in red, so you would show quite well when a quadratic model improves its respective linear model.

Reviewer #2

(Remarks to the Author)

Thanks for the updating manuscript according to comments.

Responses to the reviewers

We would like to thank both reviewers very much for their valuable comments, which helped us to
substantially improve the manuscript.

Reviewer #1 (Remarks to the Author):

Dear Editor,

I have carefully revised the manuscript titled “Unimodal relationships between plant biomass and the
environment in global grasslands”.

Here, the authors analysed a global grassland database and tested unimodal and interactive
relationships between aboveground biomass and some of its potential drivers.
In overall, I think the authors have addressed a relevant topic in ecology, and the conclusions of the
study are important for the field and for a broader audience (e.g., modeler of global vegetation and
carbon). Maybe the authors should highlight this a little bit more. In my personal view, I agree with the
authors, curves and interactions are often neglected in ecology, so we must insist they are important,
and provide evidence. We thank the reviewer very much for the detailed and insightful comments on
our manuscript, which helped us to substantially improve the manuscript.

We highlighted the importance of the study a little bit more at the end of the first paragraph of the
Introduction by pointing out that interactions in ecology are often assumed but rarely tested despite
the fact that interactions are very likely important for modelling relationships in grasslands at the
global scale (lines 84 to 86).

However, I see some problems, important ones in my opinion, so I would suggest a deep revision.
These problems are related with the concrete questions of the study and the statistical analyses
chosen. I will try to explain how, in my view, this is a major problem as it affects to the global coherence
of the paper.

First, and most important, the authors carried out a piecewise SEM model, with some interesting
results. However, it does not fit with any of the concrete questions stated at the end of the
introduction. Then, we don't know which hypothesis were tested by the authors with this analysis.
We use the SEM in to explore contributions of multiple variables to explain variability in APB. This is
important in our analysis because of co-linearity of various variables, for instance soil nitrogen and
MAP. We admit that it was not entirely clear from our hypotheses and research questions why we
conducted this analysis. In order to motivate the analysis more explicitly, we added a fourth research
question, which reads as follows: “4) To which extent do different variables (that are related with APB
through quadratic functions) explain variation in APB when considered together?” (lines 151/152). In
addition, we explain in the last two sentences of the Introduction that the SEM analysis was conducted
to explore contributions of multiple variables to explain variability in APB (lines 162/163).

Second, the authors argue in the methods that they “aggregated the data at the site level because the
climate variables vary among the sites but not among the plots of one site. In addition, different plots
of one site are not independent of each other (which is a pre-requisite for regression analysis), which
is why we conducted regression analyses based on the plot-level data” (L 434-438). I disagree with
their decision, as there are other well-known specific options to deal with this type of data, like mixed
linear models (piecewise SEM models and other SEM options allow to include random effects too). I
think authors should, at least, to include a convincing justification of why they chose this procedure
and not mixed linear effects models. I want to highlight that, by aggregating the data at the site level,
the sample size is reduced, so is the power of the analysis. Note that in the piecewise SEM model, some
variables like soil K or P were discarded for having only marginal, but not significant effects. I think this

is a problem because in the discussion the authors posit these variables could actually have an effect
on aboveground biomass (L 252). We agree with the reviewer that it is good to provide a more detailed
justification of why we aggregated the data at the site level.

The aim of our study is to understand how plant biomass is related to abiotic factors and plant diversity
in grasslands spanning a wide range of climate conditions. As a consequence, the scale of inference of
our study is the global scale, and not the local scale of a single site. Since the climate variables (mean
annual precipitation, mean annual temperature, and the aridity index, etc.) differ among sites, but not
within a site (i.e., in an area of about $\sim 1015 \text{ m}^2$), we aggregated the data at the site scale by calculating
means across all plots of one site. Aggregating plot-level data at the site-scale is common practice in
global studies in ecology (e.g., Adler et al., 2011).

Our approach is based on the common understanding that variability at different spatial scales is
caused by different drivers (Levin, 1992; Wiesmeier et al., 2019), which implies that the analysis of
small-scale variability has very limited value for understanding variability occurring at the large scale.
We calculated the arithmetic means of the plots at each site (called site-means in the following), and
conducted regression analyses based on these site-means. Computing the mean of about 30 plots per
site allowed us to get a representative estimate of soil chemical properties as well as plant species
diversity and plant biomass.

We aggregate the data at the site level because, first, we are interested in the global scale and
variability at the small scale, (among plots located at one site) is usually irrelevant for understanding
differences at the global scale. Second, the climate variables do not vary at the site-scale, and third,
the measurements at different plots of one site are not independent of each other, which is the pre-
requisite for a regression analysis.

We acknowledge the variability that exists at each site, and we presented the data measured at each
plot for all sites in Fig. 2 to give insight into the intra-site variability. However, variability at the small
scale, i.e., among plots, is usually irrelevant for understanding variability among grasslands at the
global scale. While studies exploring why biomass varies within sites are certainly valuable this is
beyond the scope of the current paper.

We added a summary of these arguments to the method section (in 'Calculations and data analyses'
lines 502-512).

In summary, I would say the manuscript in general needs to improve the link of the introduction and
discussion sections with the statistical analyses and their results. It is quite consistent but sometimes
there are important gaps. These can be filled by both improving the introduction and the discussion,
as well as the statistical approach and/or its justification. We improved the manuscript according to
the reviewer's comments. Specifically, we increased the connection between Introduction/Discussion
and the statistical analyses, for instance by adding a fourth research question that motivates the use
of SEM (lines 151/152) and by adding more detailed climate and plant diversity variables to the analysis
(see detailed answers above and below).

I also have some doubts about the choice of the Shannon's index as a proxy of plant biodiversity, that
I have explained in the detailed comments. In the revised manuscript, we included the Simpson index,
species richness, and evenness (besides the Shannon index, which was already part of the study) in the
analyses. We choose the Shannon and the Simpson index as two contrasting measures of diversity.
The first is more sensitive to differences in rare species abundance, while the latter is more sensitive
to differences in the most abundant species (Peet, 1974). The results of the analyses involving these
different measures of biodiversity are given in Table 1 and Table S1. In addition, an explanation of the
choice of these measures was added in the Methods section (lines 497 to 500).

Finally, I would suggest the authors to test more climate and plant diversity variables and to change
the title to something less descriptive and more eye-catching. We changed the title of the manuscript
(see next answer). In addition, we included more climate and plant diversity variables. We already
commented on plant diversity variables in the previous answer. We included more climate variables in
the regression analysis, and the results are shown in Tables 1, 2, and S1. Besides data on mean annual
precipitation (MAP) and mean annual temperature (MAT), we used mean precipitation of the wettest
quarter of the year (PWet), mean temperature of the driest quarter of the year (TDry), mean
temperature of warmest quarter of the year (TWarm), and maximum temperature of warmest month
(TMax).

In the lines bellow, I include more detailed comments. I hope they are useful to improve the
manuscript:

**Title:**

You should change the title since not everybody who looks for curve relationships is going to look for
the word "unimodal". also, you have found much more than that. You should look for a more eye-
catching title. We changed the title to "*Complex relationships between plant biomass, abiotic factors,
and plant diversity in global grasslands*".

**Introduction:**

In general, its quite well exposed, but I think it's sometimes incomplete (you don't mention MAT until
the line 115, and quite briefly if you compare with other variables). I am not saying you should dedicate
the same text to all variables, but in general I find the explanation about some variables is somehow
scarce. We added more details about the relationships between APB and temperature (lines 122 to
124) as well as APB and nitrogen (lines 110 to 115 and lines 133 to 135) in the Introduction. At the
same time, we tried to keep the Introduction focused and concentrated on the variables for which a
unimodal (curved) relationship with plant biomass can be expected.

L 83: Here I would suggest to remark this is a common problem with interactions (an unimodal
relationships) in ecology "widely assumed, but rarely tested" (Mantyka-pringle et ql. 2012). Thats why
this paper is important! Also, I would highlight the connection with vegetation and soil models as you
do in the conclusions.

See the following references:

*Mantyka-pringle, C. S., Martin, T. G., & Rhodes, J. R. (2012). Interactions between climate and habitat
loss effects on biodiversity: A systematic review and meta-analysis. Global Change Biology, 18(4),
1239–1252. <https://doi.org/10.1111/j.1365-2486.2011.02593.x>*

*Rillig, Matthias C., Masahiro Ryo, Anika Lehmann, Carlos A. Aguilar-Trigueros, Sabine Buchert, Anja
Wulf, Aiko Iwasaki, Julien Roy, and Gaowen Yang (2019), The role of multiple global change factors in
driving soil functions and microbial biodiversity." Science, 366, 886{890, doi: 10.1126/science.aay2832.
Spiegelberger, Thomas, Fran cois Gillet, Bernard Amiaud, Aurèlie Thiebault, Pierre Mariotte, and
Alexandre Buttler (2012), How do plant community ecologists consider the complementarity of
observational, experimental and theoretical modelling approaches?" Plant Ecology and Evolution, 145,
4 doi:10.5091/plecevo.2012.699.*

*Sirami, Clelia, Paul Caplat, Simon Popy, Alex Clamens, Raphael Arlettaz, Friedreric Jiguet, Lluís Brotons,
and Jean Louis Martin (2017), Impacts of global change on species distributions: obstacles and solutions
to integrate climate and land.*

We thank the reviewer for the suggestions and the recommended literature. We added a sentence
about the importance of testing for interactions at the end of the first paragraph of the Introduction,
as suggested, and added some of the recommended references (lines 84 to 86).

L 95: This is quite good hypothesis and is quite well exposed. Thanks!

L 102 In this case, I would expect some interactions with precipitation. Yes, we agree. In the revised
version of the manuscript, we explain this with more detail in the following paragraph (lines 116-124).

L 108: Incomplete if we compare to the previous variables (see pH). I think this is important, since for
pH is quite well-known but for nitrogen is not so clear. I think you develop something about soil N
bellow. Maybe you should mention it here instead. We added more details about nitrogen in the
Introduction (lines 110 to 115 and lines 133 to 135), as suggested.

L 114 Then we don't expect unimodal relationships (or not only) but interactions. Maybe that should
be mentioned at the beginning of the paragraph. Actually, an unimodal relationship its an interaction
of an explanatory variable with itself. Yes, we agree. In the Introduction, first unimodal relationships
between APB and other environmental factors are tackled (lines 87 to 115), and then, in a separate
paragraph, interactions among different variables (lines 116 to 124). Since this structure was
apparently not entirely clear, we reworded one sentence of the first paragraph of the Introduction,
and added an introductory sentence to the paragraph about interactions that indicates more clearly
that the topic of this paragraph are interactions.

L 112 Good hypothesis again. Thanks!

L 125: Here it is, the soil N. Anyway, I miss some explanations about why the aboveground biomass
should change with soil N. By the way, the authors focus only on total N (which is mostly organic N in
natural grasslands). When I read the first mention to soil N in line 108, I wrote: "I will wait also some
interaction with plant diversity, as the effects of temporal and spatial niche complementarity of species
(e.g. grasses and legumes, but also forbs) in nutrient acquisition are widely (but not completely)
studied". Maybe some phrases similar to this one could help the readers. We agree. Interactions of
nitrogen and plant diversity are tackled in the section about diversity, in the last paragraph of the
Introduction (lines 132 to 135).

L 130 This is also known, interactions are widely assumed, but rarely tested, as I mentioned above. This
is why I would suggest the authors to include a table with some recent or classical works addressing
the main drivers of aboveground plant biomass at wide (from regional to global) scales with climate
ranges, a brief list of the variables tested, and if they tested interactions or not. I know this implies
some work, but also would be a powerful way to transmit the reader the importance of this study. We
agree that it would be interesting to conduct a meta-analysis on the main drivers of aboveground plant
biomass at different scales based on published studies. However, given the large number of studies
about this topic, we think that this exceeds the scope (and the word limit) of this article.

L 137 I would definitively include temperature seasonality (maybe the relationship is unimodal too)
and precipitation on the warmest quarter (i.e. mean summer precipitation) at least. See my comments
to the methods. We followed the advice of the reviewer and included different measures of
temperature (besides MAT, which was already included in the analysis) in the regression analysis.
These are mean precipitation in the wettest quarter of the year, mean temperature of the driest
quarter of the year, mean temperature of warmest quarter of the year, and maximum temperature of
warmest month.

L 141 Did you try with climate variables? Yes, we looked into this. There is no significant relationship
between plant diversity and different climate variables in their relationships with APB in this dataset.
In order to maintain this study focused, we concentrate the study on the test of hypotheses derived
from the literature (rather than exploring all possible interactions).

L 143 Then you performed a test/model that does not answer any of your questions (the piecewise
SEM). I assume is because you are interested in provide evidence for causal models, you want to show
direct and indirect effects of some variables (MAP), and of course, to put all the variables together
(before you tested only individual correlations). This is important because is the only way to justify
your method based of pairwise correlations instead of working directly with a single linear model that
includes all predictors as candidate explanatory variables. This last approach could discard any variable
(E.G MAP) that could affect AB biomass thought other variables. *We use the SEM to explore co-*
*contributions of multiple variables to explain variability in APB. This is important in our analysis*
*because of co-linearity of various variables, for instance soil nitrogen and MAP. We admit that it was*
*not entirely clear from our hypotheses and research questions why we chose this approach. Therefore,*
*we added a fourth research question: “4) To which extent do different variables (that are related with*
*APB through quadratic functions) explain variation in APB when considered together?” (lines 151/152)*

L 144 Were this data used before for other papers? if yes, it is ok, but mention then. *Most of the data*
*about the 116 sites has not been published before. We added the information that the data has been*
*collected by the Nutrient Network at the end of the Introduction (lines 156/157).*

**Results:**

I think they are well exposed. *Thanks!*

L 153: You can say the quadratic model performed better ($\Delta AIC > 2$) and that would be enough
(Burnham and Anderson 2002). *Yes, we shortened this.*

L 163 I think it is a good idea to write the number of samples ($n = 116$ or 55 , together with the P and
the R^2). So that is easy for the reader as they don't have to remember which analyses were done with
each set. *We state in the text whether the analysis refers to the 55 or to the 116 sites, and we always*
*refer to Table 1 where this information is very clearly displayed. Adding more numbers to the text will*
*decrease its readability, and the numbers are already in the table.*

L 173 Table 1, you miss a space between Table and 1. *Corrected*

L 193 This seems unnecessary to me. Actually, you transformed the variable to avoid error I or II types,
that can come from de deviation of the normality of your variable/residuals. *We did this additional*
*analysis because of a paper about interactions in statistical models (Duncan and Kefford, 2021) which*
*shows that in some cases, an interaction can result from the log-transformation. In order to show that*
*this is not the case here, we conducted the regression analysis also for the non-transformed data. The*
*results show that interactions are statistically significant independent of whether the analysis was*
*conducted with the log-transformed data or the untransformed data (see Table S2). The log-*
*transformation was necessary for statistical reasons (distribution of residuals and heteroscedasticity).*
*In the revised manuscript, we added an explanation together with a reference to this paper (lines 545*
*to 549).*

L 202 I think this suggest MATxMAP does not affect APM directly, but through other variables. This is
what you test in a SEM. I think you should, at least, explain if you can check this or not. *The interaction*
*MAT*MAP is part of the model presented in Fig. 5 (see grey box indicating the interaction). We*
*included the following sentences in the methods section about an additional version of the model in*
*which not only APB but also soil nitrogen and clay are affected by this interaction. “In addition, we*
*tested a version of this model in which soil nitrogen and clay are not only affected by MAP and MAT*
*(without interaction) but also by the interaction of the two variables. However, this model had a higher*
*AIC than the model in which soil nitrogen and clay are affected by MAP and MAT without interaction.”*

In addition, we explain in the Discussion that single variables or interactions (of variables) that are
significantly related with the dependent variable when considered in isolation might not be
significantly related to the dependent variable in a more complex model. This is because the variability
of the dependent variable might be captured to a larger extent by other variables (lines 329 to 334).

**Discussion:**

As I have already mentioned, I would say the manuscript in general needs to improve the link of the
introduction and discussion sections with statistical analyses and the results. Here I provide some
comments/suggestions with the hope they help the authors to improve the discussion.

We added a fourth research questions to the Introduction to motivate the SEM analysis more explicitly
(see also our very first answer above).

229 This is a common problem in ecological papers. See
*Maire, V., Wright, I. J., Prentice, I. C., Batjes, N. H., Bhaskar, R., van Bodegom, P. M., Cornwell, W. K.,*
*Ellsworth, D., Niinemets, Ü., Ordonez, A., Reich, P. B., & Santiago, L. S. (2015). Global effects of soil and*
*climate on leaf photosynthetic traits and rates. *Global Ecology and Biogeography*, 24(6), 706–717.*
<https://doi.org/10.1111/geb.12296>

Yes, we agree.

L 239 I think this could be clearer in the writing. Also, you specify the temperature but not the
precipitation limits. We added the MAP range according to the Whittaker classification, which
emphasizes that some of the grasslands analyzed here are located in areas that according to this
classification would naturally be covered by forest (line 268).

The ranges of some of the abiotic variables are given in the beginning of the Materials and Methods
section. We added some information about this at the end of the Introduction (lines 155 to 157).

L 245 I think you want to write “Consequently, not “furthermore.” Changed as suggested.

L 260 You only talk about nitrogen bellow. The point on potassium is interesting, because as it is not
part of organic structures it’s cycle has differences with N and other nutrients.
*McGrath, Joshua, John Spargo, and Chad Penn (2014), Soil Fertility and Plant Nutrition, 166{184.*
*Elsevier, doi: 10.1016/B978-0-444-52512-3.00249-7.*

*Sardans, Jordi and Josep Penuelas (2015), Potassium: A neglected nutrient in global change. doi:*
*10.1111/geb.12259.*

I think you could reinforce your interpretation by testing K relationships too, and include it in the
piecewise model. Otherwise, you should mention why not. I understand potassium was discarded in
the piecewise SEM model. This is why I think mixed models are more appropriate for your dataset than
site aggregation. Potassium is included in the correlation analysis (Fig. 3). We discuss the relationship
between potassium and precipitation in the paragraph referred to here (lines 288 to 293), and we
come back to potassium at the end of the Discussion in the last paragraph before the Conclusion,
where we discuss its relationship with clay (lines 424 to 431).

We have no reason to hypothesize that there is a unimodal relationship between APB and potassium.
Therefore, we had not included potassium in the (unimodal) regression analysis (Table 1 and 2).
However, we changed this in the revised version of the manuscript, following the reviewer’s
suggestion. In order to be consistent, we also added calcium to this analysis, and we found a significant
linear relationship between ABP and calcium but not ABP and potassium (Table 1). Furthermore, we
included the second reference suggested by the reviewer in the Discussion.

L 261 And you did not find interaction between MAP and Clay, did you? I would mention that, because
that as it reinforces your argument. Interactions between MAP and Clay are common in literature, by
the way:

*Rodríguez, A., Canals, R. M., Plaixats, J., Albanell, E., Debouk, H., Garcia-Pausas, J., San Emeterio, L.,*
*Ribas, À., Jimenez, J. J., & Sebastià, M.-T. T. (2020). Interactions between biogeochemical and*
*management factors explain soil organic carbon in Pyrenean grasslands. Biogeosciences, 17(23), 6033–*
*6050. <https://doi.org/10.5194/bg-17-6033-2020>*

and references therein. We added that there was no significant interaction among clay and MAP and
referred to the corresponding table (Table S1), in this part of the Discussion. We also included the
reference in the Introduction.

L 283 Is there no way to represent the MAT*MAP interaction in your piecewise model? Maybe checking
the relative importance of MAP and MAT and its interaction (e.g. relaimpo package) could give us some
light. Or to include the interaction but not affecting directly to plant biomass. The interaction
MAT*MAP is/was part of the SEM (see gray box). For details, see our answer to the comment about
line 202 (above).

L 302 Is it possible to include a SEM for the 116 sites, only for the variables that are available? at least
to put in the supp matts and compare. We would not know how to do this in a meaningful way. Soil
nitrogen turned out be the variable that explains the largest proportion of the variability of APB.
However, for most of the 116 sites, we do not have data on soil nitrogen.

306 Logical, soil variables will affect the vegetation variables (including biomass) more that climate
variables, especially if these are taken from a map (Rodríguez et al. 2022 found the same for plant trait
indices)

*Rodríguez, A., de Vries, F. T., Manning, P., Sebastià, M. T., & Bardgett, R. D. (2022). Soil Abiotic*
*Properties Shape Plant Functional Diversity Across Temperate Grassland Plant Communities.*
*Ecosystems. <https://doi.org/10.1007/s10021-022-00812-2>*

Whether climate or soil variables dominate is likely scale-dependent. At the small to intermediate
spatial scale (say region to country, as in the mentioned reference) soil factors are likely more
important than climate factors because the latter do not vary so much. However, at the global scale,
climate factors are likely more important than at the intermediate scale.

L 311 Could you provide some explanation? For instance, because soil total nitrogen is not a perfect
indicator of nitrogen availability. Most N is immobilised and then the available N probably does not
satisfy the whole plant nitrogen demand. But maybe you can find a better one. We added the
explanation that atmospheric nitrogen deposition likely contributed to plant nitrogen nutrition (lines
266 to 267).

L 319. What about Richness, Simpson, evenness? See comments in methods section. We added species
richness, evenness, and the Simpson index to the analysis (see Table 1 and Table S1).

L 334 This is quite important, the dominance effect primes in high nutrient abundance scenarios, and
the niche complementarity in the low nutrient scenarios. Then I suggest you to discuss if this is
consistent with studies about functional diversity and plant biomass, because most functional diversity
indices are actually hypotheses of one of these effects (e.g. CWMs for dominance effects and Rao's Q
for niche complementarity). We added a sentence saying that a high plant species diversity that results
in a high functional diversity in the use of chemical nitrogen forms might be beneficial for APB under
low nitrogen availability (lines 393 to 396).

L 339: Did these authors check for the interaction, or only the main effect? If your results are a novelty,
you should highlight it. No, these studies did not explore (explicitly) the interaction between plant
species diversity and nitrogen. We try to stress a little bit more the novelty of our study, in the revised
version of the manuscript (lines 399/400).

L 370 This is very accurate. Sometimes this paper has very high quality and interesting details. Thanks.

L 344 or other nutrients. Yes. We replaced “nitrogen” by “nutrient”.

**Conclusion**

Maybe the authors want to change something after the revision. We checked carefully that all new
results are in line with the conclusions, but we did not make changes to the Conclusion section (since
the new results are in line with the conclusions).

**Methods:**

I think they are very well explained, though as I mentioned, I disagree with the site aggregation and
with the way of choosing a plant diversity index. See detailed answers below.

L 421 Were is pH? Yes, there is data on pH for all 55 sites. This was an unclear phrase and we improved
it.

L 423 From Worldclim or Worldclim 2.0 (which is the same approach, but the second was built with
much more sites, and is the one you must use). This database has 18 climatic variables, so there are
some possibilities to test other climate variables. I would encourage you to test temperature or
precipitation seasonality, for instance, as these are the same case as interactions or unimodal effects,
they are not tested very often. Yes, we used Worldclim 2.0 (we added this information). We included
more climate variables in the regression analysis (Tables 1, 2, and S1). Besides data on mean annual
precipitation (MAP) and mean annual temperature (MAT), we used mean precipitation of the wettest
quarter of the year (PWet), mean temperature of the driest quarter of the year (TDry), mean
temperature of warmest quarter of the year (TWarm), and maximum temperature of warmest month
(TMax).

L 429 Why Shannon’s diversity is used? I miss some justification. I assume is because the authors want
to summarise richness and evenness together (number of species and their proportions). However,
using both richness and the Camargo’s or Pielou’s evenness (you can choose the one that less
correlates with richness) instead of just one index would be more informative than using a composite
index. They could even test which approach is best for describing the relationship between plant
diversity and aboveground productivity: richness and evenness, Shannon or Simpson. Anyway,
sometimes Shannon index is highly correlated with species richness, that’s why I think they should
justify this. Note that you calculate Shannon’s index with a package that explains Simpson’s index could
be preferable: typing “?diversity” in R: "Better stories can be told about Simpson's index than about
Shannon's index, and still grander narratives about rarefaction (Hurlbert 1971). However, these indices
are all very closely related (Hill 1973), and there is no reason to despise one more than others... In
particular, the exponent of the Shannon index is linearly related to inverse Simpson (Hill 1973)
although the former may be more sensitive to rare species).

For justifying your decision, you can check also

Roswell, M., Dushoff, J., & Winfree, R. (2021). A conceptual guide to measuring species diversity. *Oikos*,
130(3), 321-338.

We agree, of course, with the statement that different diversity indexes differ in their sensitivity to
rare and common species. Following the reviewer’s advice, we therefore widened the analysis, and
included the Simpson index, species richness, and evenness in the analysis. We choose the Shannon

and the Simpson's index as two contrasting measures of diversity. The first is more sensitive to
differences in rare species abundance, while the latter is more sensitive to difference in the most
abundant species (Peet, 1974). In the revision, we included the Simpson index, species richness and
evenness (for results see Tables 1 and S1), and added an explanation for the choice of these two
specific diversity indexes to the methods section. We added the following explanation. "*We choose*
*the Shannon and the Simpson index as two contrasting measures of diversity. The first is more sensitive*
*to differences in rare species abundance, while the latter is more sensitive to difference in the most*
*abundant species (Peet, 1974).*" (lines 497 to 500).

The regressions with only one explanatory variable showed that none of the measures of plant
diversity, species numbers or evenness was significantly related to plant biomass (neither through a
linear nor through a unimodal function). Furthermore, the regression analyses with two explanatory
variable showed that the results were very similar independent of which measure was used (see Table
S1).

L 444 you tested the normality of residuals, but you also should provide some evidence of their
homoscedasticity, as heteroscedasticity could induce type I error in your models. Also, some
information about the influence of outliers in the model estimates should be advisable (e.g. a plot of
Cook's distance and Leverage). We created scatterplots that show the fitted values of the model vs.
the residuals of those fitted values to evaluate heteroscedasticity, and we added this information to
the Method section (line517/518) . In addition, we plotted all significant regressions in order to verify
that the regressions are not based on outliers.

L 447 This is unclear. I guess you mean you fit the equations in this order (from less complex to more
complex) and you accept the more complex equation if it improves the previous one when the AIC
difference is higher than two. The order is actually not relevant. Relevant for the comparison is the
AIC, as explained in the manuscript. The summaries of all four models that were calculated for each
pair of predictors are shown in Table S1, and the reader can compare the model summaries.

L 547 In my view, the threshold should be based on R square too (or in variance inflation factor, see the
HH package). In you base on p value, maybe you are discarding some interactions just because the
correlation is significant but weak (e.g. R squ = 0.1 or 0.2 can give a p-value < 0.05 and I think you can
still test the interaction with no problem). This comment is not entirely clear (and the line number
seems incorrect). We calculated all four models (equations 4 to 6). The P values and the P values of the
interaction terms as well as the multiple R², the adjusted R², and the AIC are displayed in Table S1. The
fact that we display all four models for each pair of predictors in Table S1 makes the model selection
very transparent for the reader.

L 461 These equations are misleading as they are slightly different to the true equations detailed one
line below. This could confound some readers. The equations are the same, we give them in the full
mathematical notation and in the R notation that many readers might be more familiar with. The R
notation is much shorter than the mathematical notation, which is very useful when referring to the
different model structures in the table (Tables 2, S1 and S2). We added a few words to indicate that
these are two different notations of the same equation (lines 533/534).

L 471, In the introduction you should explain what do you aim with this analysis. We added a fourth
research question in the Introduction, which motivates the use of SEM more clearly (for more details
see our second answer above).

L 481 In supplementary material I suggest you to include some diagrams (maybe just with the diagrams
and the AIC, or a table of AICs in addition) to give an idea of how was your model selection. The

summery of all four models is shown for all pairs of predictors in Table S1, and the model with the
lowest AIC is marked in bold font. This is the selected model that is also shown in the manuscript (Table
2). By displaying the whole summary of all models in the Supplement, we keep the model selection
very transparent for the reader.

L 492 From my point of view, this is a problem, because you explain things about available potassium
in the discussion, without mentioning that it is not significant and out of your final model, and you did
not mention this. These are two different things. Potassium is not significantly correlated with APB
(see Table 1). However, it is significantly correlated with clay and MAP (see Figure 3) and this is what
we discuss here.

**Tables**

Table 1:, maybe you should move the information for all sites analyses to supplementary material, to
say your results were consistent, and focus on the 55 sites. Where is potassium? We would like to
keep the analysis for the 116 sites in Table 1. While we do not have soil data for all 116 sites, the
analysis of the relationship between APB and climate variables across all 116 sites is still very valuable
for this study, and therefore we would not like to move it to the Supplement. It is also beneficial to
show that the results for the large dataset and the smaller subset are very similar.

We included the models for APB as a function of potassium and (in order to be consistent) calcium in
Table 1. The reason why they were not in the Table before was that we wanted to keep the study more
strongly focused on the hypotheses and research questions, and therefore we had initially only
included variables in the regression analysis for which we have indications (from literature or theory)
that their relationship with APB follows a unimodal function. However, we changed this now.

Table 2: I would write delta AIC respect to the model of reference in the previous table. Otherwise, AIC
itself is meaningless. Yes, we agree that the AIC in Table 2 is rather meaningless. However, in Table S1
the AIC is essential because we compared the models shown in Table S1 based on the AIC. We removed
the AIC from Table 2 but kept the AIC in Table S1.

Figure 2: I know is in the table, but I suggest to mention the R² and the p-value in the figure or in the
caption. We added the R² and p values in Fig. 2, as suggested.

Figure 5: the mention of AIC in the box is meaningless, unless you have a table of different SEM models
with different AICs. We followed the reviewer's advice and removed the AIC from Fig. 5.

**Reviewer #2 (Remarks to the Author):**

This study explored the relationship between grassland production and environmental interactions
using data from 116 grasslands on six continents. The study found that plant biomass has unimodal
relationships with factors like mean annual precipitation (MAP) and soil nitrogen. Plant diversity and
biomass are positively related at low nitrogen levels and negatively at high levels.

The data collection was done between 2007 and 2020, it needed to be clearly mentioned which month
or mowing frequency/grazing condition or what stage of the grass growth. Aboveground plant biomass
was measured at peak biomass, i.e., at the specific time of the year when aboveground plant biomass
is largest. This was done by local grassland ecologist who are familiar with the site and its phenology.
This information was given in the Methods section. In the revised version of the manuscript, we added
this information at the very end of the Introduction (lines 158 to 160).

The authors discussed the impact of abiotic factors on grass biomass. However, the reasons for
avoiding elevation and slope should have been mentioned or tried to be used in the modelling process.
Following the reviewer's suggestion, we included elevation of the grasslands in the analyses. The
results of this are shown in Table 1 and Figure 3. The reason why we had not included elevation before
is that elevation itself cannot directly affect plant biomass, but only indirectly through temperature or
precipitation, etc. In the revised version of the manuscript, we discuss the relationship between
elevation and APB in the second paragraph of the Discussion section (lines 273 to 275).

Further land use type (e.g., agriculture, nature protected, grazing) and land use intensity (intensively
or extensively managed, fertilised or not) could change plant biomass available. Yet needed to be more
information could not be found in the manuscript about those factors.
Consideration should be given to the potential of growing degree days as a more effective method for
establishing the relationship between biomass and temperature, as opposed to MAT. The grassland
sites did not receive any fertilizer and were not experimentally manipulated at the time of study. This
information was/is in the method section, and in the revised version of the manuscript, we added it at
also the end of the Introduction (lines 157/158).

**References**

- Adler, P. B., Seabloom, E. W., Borer, E. T., Hillebrand, H., Hautier, Y., Hector, A., ... & Yang, L. H. (2011).
Productivity is a poor predictor of plant species richness. *Science*, 333(6050), 1750-1753.
- Duncan, R. P., & Kefford, B. J. (2021). Interactions in statistical models: three things to know. *Methods*
*in Ecology and Evolution*, 12(12), 2287-2297.
- Levin, S. A. (1992). The problem of pattern and scale in ecology: the Robert H. MacArthur award
lecture. *Ecology*, 73(6), 1943-1967.
- Peet, R. K. (1974). The measurement of species diversity. *Annual Review of Ecology and Systematics*,
285-307.
- Wiesmeier, M., Urbanski, L., Hobbey, E., Lang, B., von Lützow, M., Marin-Spiotta, E., ... & Kögel-Knabner,
I. (2019). Soil organic carbon storage as a key function of soils-A review of drivers and indicators
at various scales. *Geoderma*, 333, 149-162.

Responses to the reviewers

We would like to thank both reviewers very much for reading the manuscript again. The comments helped us to further improve the manuscript.

Reviewer #1 (Remarks to the Author):

Dear Editor,

I have carefully revised the manuscript and the authors answer to mi revision. In general, they answered satisfactorily to my comments. I think the changes they made have improved the manuscript, especially the discussion section, which seems now quite convincing and transmit the ideas in such an efficient way.

I am not totally convinced about the title, that now seems somehow vague. I would suggest something more explicit, like “The importance of non-linear and interactive relationships between plant diversity and abiotic factors for plant biomass in global grasslands”. Maybe the editor or the authors prefer the current title or want to suggest another one.

In mathematics, quadratic models belong to the class of linear models. Therefore, the manuscript refers to “quadratic, unimodal models” versus “linear, monotonic models” (but we do not use the term “non-linear” when referring to the quadratic models). For the same reason, we do not want to use the term “non-linear” in the title (as the reviewer suggested). However, since the reviewer found the title of the manuscript vague, we improved it by replacing “complex” by “interactive and unimodal”. The full title of the manuscript now reads, *“Interactive and unimodal relationships between plant biomass, abiotic factors, and plant diversity in global grasslands”*

I also think that the last research question they added is incomplete, as they use a structural equation model, then “to which extent do different variables (that are related with APB through quadratic functions) explain variation in APB when considered together?” is not only what they test. They could have used a multiple linear model for this purpose. However, their election of SEM is accurate because with a conventional linear model they would take the risk of discarding some variables that are related or have causal relationships between them. A good example in this case is MAT and Clay. I agree this is implicit in the manuscript but should be made explicit in the different sections (introduction, methods, in the discussion, why not...) at least briefly. You could take a look at the “Hierarchy of controls over function” hypothesis exposed here:

De Vries, F. T., Manning, P., Tallwin, J. R., Mortimer, S. R., Pilgrim, E. S., Harrison, K. A., ... & Bardgett, R. D. (2012). Abiotic drivers and plant traits explain landscape-scale patterns in soil microbial communities. Ecology letters, 15(11), 1230-1239.

Maybe the ideas there can help. My apologies if I was not clear enough about these two last points in my previous revision. Anyway, I insist in the great improvement of the manuscript. We improved the fourth research question. It now reads, *“To what extent do different variables (that are related with APB through different functions, covary or interact with each other) explain variation in APB when considered together?”*

Additionally, after the changes you have done, I believe that the end of the discussion, you could take profit on your SEM to finish with some comments of the global picture you found in your paper, and that this is highly relevant for improving vegetation and, why not, carbon models too. A short paragraph with a few references should be enough. You could even use this idea to improve your last question at the introduction, as in the rest of your paper you test a number of ABP drivers one by one (or in paris, in some cases), but the idea of the paper is to propose improvements in biomass and carbon models, and these consider multiple variables as you do with your SEM. We included the following sentences at the end of the Discussion, just before the Conclusion: *“Our study demonstrates important interactions between abiotic and biotic variables in global grasslands. Specifically, our results demonstrate that plant species diversity and nitrogen interact significantly in their relationships with APB (Table 2 and Fig. 4). Furthermore, the study shows that MAT is not significantly related with APB (Table 1), but with*

the interaction of MAT and MAP (Table 2). Our results highlight the importance to account for the interactive and unimodal relationships between plant biomass and several environmental variables when analyzing and modelling plant biomass at the global scale. Including interactive and unimodal relationships in global vegetation and carbon models likely improves their ability to predict primary production at the global scale.

Concerning the SEM, something maybe I would mention more explicitly is that the indirect effects of MAP seemed to be linear, but the direct effects are unimodal. We included the following sentence on page 14. *“The SEM shows that the direct effect of MAP on APB follows a quadratic function (see black asterisks and letters next to the quadratic relationship in Fig. 5), while MAP affects APB also indirectly through its linear effect on soil nitrogen (Fig. 5).”*

I would also like to suggest some minor changes, in case the authors want to consider them. L. 116. I wrote that in my previous revision, but maybe it's more scientific to say "have multiplicative effects with itself". I usually use the phrase “in unimodal relationships, in which one explanatory variable interacts with itself” to try to help people to understand, but inter-action implies minimum 2 actors or components. We changed this to *“quadratic models, in which a variable is multiplied with itself”*.

L 124. Maybe a ref. here... We included a reference.

L 151 I have already exposed my view about this question above. See our reply above

L163 Also, see above. This is not the only thing you are doing with a SEM, and in fact, it is well done, so a better explanation would put into value the work done. We improved this sentence. It now reads, *“Structural equation modelling allowed us to explore the contributions of multiple variables (which are related with APB through different functions, covary or interact with each other) to explain variability in APB.”*

L 242 “APB and MAP followed a quadratic model.” Maybe, they follow a quadratic relationship? Not a model. We believe that “follow a quadratic model” is more accurate than “follow a quadratic relationship”.

L 426: To be honest, I did not understand if in Fig. 4B Clay is represented linearly or in an unimodal relationship. Checking table 1, I see the quadratic model its significant, but the AIC is higher than the linear model. Maybe it would be clearer to say the quadratic relationship of clay is significant but negligible in terms of size, then the relation is mostly linearly positive, as you expose. This should be clear from the caption of Fig. 4 and Table 1. The caption of Fig. 4 says “For model summaries see Table 2”. Table 2 shows that the model of Clay and MAP is quadratic for MAP and linear for Clay (without interaction of the two variables). For model comparison see Table S1.

L. 430-433 the second reason for using SEM... This comment is not entirely clear to us.

Table 1: I don't see the relevance of showing adjusted R² here, since this is a correction for a high number of variables. They are very similar because you only have 1 or 2 terms in the linear or quadratic model, respectively. Also, maybe you want to simplify this table by suppressing AIC in the linear model table and replacing AIC by Delta AIC in the quadratic model, being Delta AIC = AICquadratic model – AIClinear model, and showing negative values in red, so you would show quite well when a quadratic model improves its respective linear model. The reviewer is, of course, right that for the models with one variable, it would be enough to only show the R². The reason why we show also the adjusted R² is that this facilitates comparisons with the models in Table 2. Furthermore, we believe that it is useful for

the reader to see the AIC and not the Delta AIC since this allows a comparison of the different models shown in Table 1 and not only a comparison of the two models (unimodal and monotonic) with the same independent variable.

Reviewer #2 (Remarks to the Author):

Thanks for the updating manuscript according to comments. We thank the reviewer for reading the manuscript again and checking the improvements.